| 1  | Joint retrieval of aerosol and water-leaving radiance from multi-spectral, multi-                                                                                                   |
|----|-------------------------------------------------------------------------------------------------------------------------------------------------------------------------------------|
| 2  | angular and polarimetric measurements over ocean                                                                                                                                    |
| 3  | Feng Xu <sup>a,</sup> *, Oleg Dubovik <sup>b</sup> , Peng-Wang Zhai <sup>c</sup> , David J. Diner <sup>a</sup> , Olga V. Kalashnikova <sup>a</sup> , Felix C. Seidel <sup>a</sup> , |
| 4  | Pavel Litvinov <sup>b</sup> , Andrii Bovchaliuk <sup>b</sup> , Michael J. Garay <sup>a</sup> , Gerard van Harten <sup>a</sup> , and Anthony B. Davis <sup>a</sup>                   |
| 5  | a): Jet Propulsion Laboratory, California Institute of Technology, Pasadena, California, USA                                                                                        |
| 6  | b): Laboratoire d'Optique Atmosphérique, UMR8518, CNRS/Universite Lille-1, Villeneuve d'Ascq, France                                                                                |
| 7  | c): Department of Physics, University of Maryland, Baltimore County, Baltimore, Maryland, USA                                                                                       |
| 8  |                                                                                                                                                                                     |
| 9  | *Corresponding author: Feng.Xu@jpl.nasa.gov                                                                                                                                         |
| 10 | Abstract                                                                                                                                                                            |
| 11 | An optimization approach has been developed for simultaneous retrieval of aerosol properties and normalized                                                                         |
| 12 | water-leaving radiance (nLw) from multi-spectral, multi-angular, and polarimetric observations over ocean. The                                                                      |
| 13 | main features of the method are (1) use of a simplified bio-optical model to estimate nLw followed by an empirical                                                                  |
| 14 | refinement within a specified range to improve its accuracy; (2) improved algorithm convergence and stability by                                                                    |
| 15 | applying constraints on the spatial smoothness of aerosol loading and Chlorophyll-a (Chl-a) concentration across                                                                    |
| 16 | neighboring image patches and spectral constraints on aerosol optical properties and on nLw across relevant bands;                                                                  |
| 17 | and (3) enhanced Jacobian calculation by modeling and storing the radiative transfer (RT) in aerosol/Rayleigh mixed                                                                 |
| 18 | layer, pure Rayleigh scattering layers, and ocean medium separately and then coupling them to calculate the field at                                                                |
| 19 | the sensor. This approach avoids unnecessary and time-consuming recalculations of RT in unperturbed layers in                                                                       |
| 20 | Jacobian evaluations. The Markov chain method is used to model RT in the aerosol/Rayleigh mixed layer and the                                                                       |
| 21 | doubling method is used for the uniform layers of the atmosphere-ocean system. Our optimization approach has                                                                        |
| 22 | been tested using radiance and polarization measurements acquired by the Airborne Multiangle SpectroPolarimetric                                                                    |
| 23 | Imager (AirMSPI) over the AERONET USC_SeaPRISM ocean site (6 February 2013) and near the AERONET La                                                                                 |
| 24 | Jolla site (14 January 2013), which respectively reported relatively high and low aerosol loadings. Validation of the                                                               |
| 25 | results is achieved through comparisons to AERONET aerosol and ocean color products and retrievals performed                                                                        |
| 26 | using the Generalized Retrieval of Aerosol and Surface Properties algorithm (Dubovik et al., 2011) on AirMSPI                                                                       |
| 27 | data. Uncertainties of aerosol and nLw retrievals due to random and systematic instrument errors are analyzed by                                                                    |

- truth-in/truth-out tests with three Chl-a concentrations, five aerosol loadings, three different types of aerosols, and
- nine combinations of solar incidence and viewing geometries.
- Keywords:
- Atmosphere and ocean system, polarized radiative transfer, aerosol retrieval, water-leaving radiance retrieval
- © 2015. All rights reserved.

#### 6 1. Introduction

Aerosols exist in the form of airborne suspensions of tiny particles that scatter and absorb sunlight, leading to significant impacts on Earth's energy and water cycles. Quantifying aerosol 8 9 influences on climate requires accurate determination of their abundances and 10 optical/microphysical properties, which are highly variable spatially and temporally. Aerosol 11 characterization is also crucial for ocean color remote sensing, as the spectral water-leaving 12 radiances account for only 10-15% of the signal observed at the top of the atmosphere (TOA) 13 and most of the signal arises from atmospheric scattering. Chlorophyll-a concentration, colored dissolved organic matter (CDOM) and other ocean optical properties retrieved from spectral 14 15 water-leaving radiance provides a measure of ocean productivity and health of ocean ecosystem. 16 Small over- or underestimates of the aerosol contribution can bias the determinations of these 17 quantities.

18 Traditional ocean color retrievals decouple the atmosphere and surface using "atmospheric 19 correction" procedures. The Ocean Biology Processing Group (OBPG) uses the atmospheric 20 correction developed by Gordon and Wang (1994) and Gordon (1997) and refined by Ahmad et 21 al. (2010). In this algorithm an aerosol optical property lookup table (LUT) is built for ten 22 aerosol models and eight relative humidity (RH) values based on the aerosol property statistics 23 from Aerosol Robotic Network (AERONET) observations (Ahmad et al., 2010). Aerosol optical 24 depth (AOD) and type are determined by fitting the observations in two near-infrared bands (748

and 869 nm), where water-leaving radiance is assumed negligible. The selected aerosol model is then extrapolated to shorter-wavelength visible bands and applied to the measured TOA radiances to retrieve normalized water-leaving radiance (nLw) (Gordon and Wang, 1994; Gordon, 1997). To reduce errors caused by this atmospheric correction procedure and instrumental radiometric uncertainties, empirical gain factors are derived by forcing agreement between retrieved nLw values and in-situ measurements obtained at the Marine Optical Buoy (MOBY) site in Lanai, Hawaii (Franz et al., 2007).

8 For single-angle, non-polarimetric instruments such as MODIS and the Sea-viewing Wide 9 Field-of-view Sensor (SeaWiFS), Franz et al. (2007) pointed out that "the performance of 10 satellite-based ocean color retrieval process is relatively insensitive to the aerosol model 11 assumption ... at least for open-ocean conditions where maritime aerosols dominate and aerosol 12 concentrations are relatively low (i.e. aerosol optical thickness generally less than 0.3 at 500 nm)." 13 Therefore, the gain factors derived from conditions at the MOBY site can be applied globally to 14 improve the agreement between satellite and in-situ nLw over deep (Case 1) waters.

15 In more challenging observing conditions, e.g., in the presence of absorbing aerosols or 16 complex, spatially diverse (Case 2) waters, imperfect knowledge of the absorbing aerosol optical 17 properties or height distribution can lead to incorrect assumptions regarding CDOM and 18 phytoplankton absorption coefficients (Moulin et al., 2001; Schollaert et al., 2003; Banzon et al., 19 2009). In addition, the vertical distribution of absorbing aerosols can affect the reflectance of the 20 ocean-atmosphere system, resulting in errors in nLw (Duforêt et al., 2007). In coastal regions, 21 where the traditional assumption of zero water-leaving radiance in the near-infrared (NIR) 22 (Gordon, 1997; Siegel et al., 2000) breaks down, backscattering from suspended hydrosol 23 particles (e.g., algae or sediment) can be misinterpreted as aerosols, leading to overestimation of

1 AOD. The resulting overcorrection can lead to underestimated or even negative water-leaving

2 radiances in the blue and green (e.g., Hu et al., 2000; Bailey et al., 2010; He et al., 2012).

The National Aeronautics and Space Administration's Pre-Aerosol, Clouds, and ocean Ecosystem (PACE) mission, with an anticipated launch date early in the next decade, is aimed at expanding upon current satellite ocean color measurements. The PACE payload is envisioned to include an ocean color spectrometer to measure ocean carbon storage and ecosystem function, and possibly a multi-angle, multi-spectral polarimeter to provide advanced data records on clouds and aerosols and to assist with atmospheric correction of the ocean biology measurements.

10 The capability of multi-angle polarimetry in characterizing aerosols for the purposes of 11 assessing their climatic or environmental impacts and improving nLw retrievals over turbid 12 waters or in the presence of absorbing (dust or carbon-containing) aerosols motivates 13 supplementing the vicarious calibration and LUT-based atmospheric correction procedures with 14 one that permits simultaneous extraction of AOD, particle properties, and nLw. Inclusion of 15 spectral bands covering the UV, visible, NIR, and shortwave infrared (SWIR), multiple view angles, and polarimetry in the retrieval enables retrieval of aerosol types that may be beyond the 16 17 capabilities of the LUT and potentially improves accuracy of both the aerosol and ocean water 18 properties. Given that measurements of atmospheric mineral dust and carbonaceous aerosols 19 show a strong spectral dependence of absorption coefficient in the near-UV (e.g., Koven and 20 Fung, 2006; Bergstrom et al., 2007; Russell et al., 2010) and have a spectral signature similar to 21 those of CDOM, accurate modeling of radiative transfer (RT) in the coupled atmosphere-ocean 22 system (CAOS) becomes necessary.

In traditional aerosol-targeted retrievals, a bio-optical model is not always necessary as the 1 2 water-leaving radiance is a small contribution to TOA signals so that it can be empirically 3 estimated or even neglected in some spectral bands. Many RT models assume a flat ocean 4 surface for specular reflection (Jin and Stamnes, 1994; Bulgarelli et al., 1999; Chami et al. 2001; Sommersten et al., 2009; Zhai et al., 2009) for simplicity of modeling. Better modeling fidelity 5 6 and accuracy can be achieved by including sea surface roughness into the RT models (Nakajima 7 and Tanaka, 1983; Fischer and Grassl, 1984; Masuda and Takashima, 1986; Kattawar and 8 Adams, 1989; Mobley, 1994; Deuzé, 1989; Jin et al., 2006; Spurr, 2006) and including the 9 water-leaving radiance and/or ocean foam reflection based on a Lambertian or a more general 10 bidirectional reflectance distribution model (Koepke, 1984; Lyapustin and Muldashev, 2001; 11 Mobley et al. 2003; Sayer et al., 2010; Sun and Lukashin, 2013; Gatebe et al., 2005). Though 12 empirical parameterization of water-leaving radiance simplifies the radiative transfer, the 13 relationship between water-leaving radiance and inherent optical properties (IOP) of dissolved or 14 suspended ocean constituents is indirect. This weakness can be overcome by using bio-optical 15 models to relate IOP directly to water-leaving radiance. This makes it feasible to perform a one-16 step retrieval of IOP and aerosol optical properties from TOA measurements of radiance and 17 polarization (e.g., Hasekamp et al. 2011), which is a complementary retrieval strategy to the 18 prevailing two-step retrieval that obtains nLw from TOA via atmospheric correction and then 19 determines IOP from nLw (IOCCG, 2006). Various RT solutions involving the use of bio-optical 20 models have been developed and can be used for this purpose. These include the invariant 21 imbedding method adopted by HydroLight (Mobley, 2008) and its faster version EcoLight 22 (Mobley, 2011a) for scalar (intensity only) RT, and the adding-doubling method (Chowdhary et

1 al., 2006) and successive-order-of-scattering method (Zhai et al., 2010) for polarized RT in the

2 CAOS.

3 Joint retrieval of aerosol and nLw properties requires supplementing the forward RT 4 calculations with a sophisticated and computationally efficient inverse model to disentangle their 5 contributions to TOA radiometry and polarimetry. Motivated by the development of a multi-6 angle imaging polarimeter at JPL-the Airborne Multiangle SpectroPolarimetric Imager 7 (AirMSPI) (Diner et al., 2013)—this paper describes the development of a coupled aerosol-ocean 8 retrieval methodology. Our method (1) employs a simplified bio-optical model to obtain a 9 reasonable estimate of nLw in the first retrieval step, followed by an empirical refinement in the 10 subsequent step; (2) applies constraints on the spatial smoothness of aerosol and Chl loadings 11 across neighboring image patches and spectral constraints on aerosol optical properties and on 12 nLw across relevant bands to improve the convergence and stability of the algorithm; and (3) 13 models and stores the RT fields in the aerosol/Rayleigh mixed layer, the pure Rayleigh scattering 14 layers, and the ocean medium separately, and then couples them to obtain the radiative field at 15 the sensor-thereby enhancing the Jacobian evaluations by reusing RT fields in the unperturbed layers. The Markov chain and doubling methods are applied to the mixed and uniform layers, 16 17 respectively, to gain computational efficiency.

The parameters of our retrieval include spectrally dependent real and imaginary parts of aerosol refractive index, aerosol concentrations of different size components, mean height and width of aerosol distribution, nonspherical particle fraction, wind speed over ocean surface, and normalized water-leaving radiance. As auxiliary product, aerosol phase matrix is obtained from the retrieved refractive index and normalized size distribution. Throughout the paper, we use the definition of "exact" normalized water-leaving radiance (nLw) given by Morel et al. (2002). It is

1 consistent with the definition adopted by Franz et al. (2007) and Zibordi et al. (2009) and is 2 related to the remote sensing reflectance ( $R_{rs}$ ) by  $R_{rs} = nLw/F_0$ , where  $F_0$  is the extraterrestrial 3 solar irradiance.

4 The paper is organized as follows. In Section 2, we introduce our development of the RT model that integrates the Markov chain and adding-doubling methods for CAOS. The multi-5 6 patch retrieval algorithm is described in Section 3. In Section 4, a truth-in/truth-out test is 7 performed to assess the retrieval uncertainties for a variety of synthetic scenarios combined from 8 three types of aerosols, five aerosol loadings, three Chl-a concentrations, three solar incidence 9 angles, four viewing geometries, and two types of measurement noise. To test the algorithm with 10 real data, retrievals applied to AirMSPI observations over the USC SeaPRISM AERONET site 11 and near the La Jolla AERONET site are compared to the independent AERONET results. A 12 summary is presented in Section 5.

## 13 2. A flexible radiative transfer model for a coupled atmosphere-ocean system

### 14 2.1 Model structure and single scattering properties

15 A five-layer model, consisting (from the bottom up) of the ocean medium, the air-water 16 interface, a pure Rayleigh layer, an aerosol/Rayleigh mixed layer, and a second pure Rayleigh 17 layer is established for the CAOS system (see Fig. 1). All layers are vertically homogeneous 18 except for the "mixed layer", where the aerosol has its own vertical distribution profile different 19 than that of the Rayleigh-scattering molecular atmosphere. The mixed layer is defined to have 20 the minimum altitude  $h_{\min}$  and maximum altitude  $h_{\max}$ . A single aerosol species is assumed to be 21 distributed throughout it with a Gaussian distribution profile characterized by mean height  $h_a$  and 22 standard deviation  $\sigma_a$  characterizing the width of the aerosol layer. Then, the aerosol 23 concentration profile  $c_a$  is

 $c_{\rm a}(h) = F_{\rm norm} \exp\left[-\frac{(h-h_{\rm a})^2}{\sigma_a^2}\right],\tag{1}$ 

where the normalization factor  $F_{\text{norm}}$  is used to ensure that  $\int_{h_{\min}}^{h_{\max}} c_k(h) dh = 1$  and evaluates to

$$F_{\text{norm}} = \frac{\sqrt{\pi}\sigma_{a}}{2} \left[ \operatorname{erf}\left(\frac{h_{\text{max}} - h_{a}}{\sigma_{a}}\right) - \operatorname{erf}\left(\frac{h_{\text{min}} - h_{a}}{\sigma_{a}}\right) \right], \quad (2)$$

where erf(x) is the error function.

Breaking the aerosol volumetric size distribution dV(r)/dln(r) into a finite number of size 6 components (Dubovik et al., 2011), the total AOD ( $\tau_a$ ) is the sum of all size components:

$$\tau_{a} = \sum_{i=1}^{N_{sc}} C_{v,i} K_{ext,a,i} = C_{v, tot} \sum_{i=1}^{N_{sc}} f_{i} K_{ext,a,i} , \qquad (3)$$

where  $N_{sc}$  is the total number of size components;  $K_{ext,a,i}$  and  $C_{v,i}$  are the extinction coefficient (in 9 units of km<sup>-1</sup>) and column volume concentration (in units of km) of the  $i^{th}$  aerosol size 10 component, respectively;  $C_{v,tot}$  is the total volume concentration ( $C_{v,tot} = C_{v,1} + C_{v,2} + C_{v,3} + ...$ ); 11 and  $f_i$  is the volume fraction of the  $i^{th}$  component ( $f_i = C_{v,i}/C_{v,tot}$ ).

Moreover, the total aerosol size distribution is constituted as

$$\frac{dV(r)}{d\ln r} = \sum_{i=1}^{N_{sc}} \frac{dV_i(r)}{d\ln r} = \sum_{i=1}^{N_{sc}} C_{v,i} \frac{dv_i(r)}{d\ln r}.$$
 (4a)

14 and the associated normalized size distribution is

$$\frac{\mathrm{d}\mathbf{v}(r)}{\mathrm{d}\ln r} = \sum_{i=1}^{N_{\mathrm{sc}}} f_i \; \frac{\mathrm{d}\mathbf{v}_i\left(r\right)}{\mathrm{d}\ln r}.$$
 (4b)

Using a log-normal volume weighted size distribution for all size components,  $dv_i(r)/dlnr$  is 17 dimensionless and is parameterized by a median radius for volume size distribution  $r_{m,i}$  and a 18 geometric standard deviation  $\sigma_i$ , namely,

$$\frac{\mathrm{d}\mathbf{v}_{i}(r)}{\mathrm{d}\ln r} = \frac{1}{\sqrt{2\pi\sigma_{i}}} \exp\left[-\frac{(\ln r - \ln r_{\mathrm{m},i})^{2}}{2\sigma_{i}^{2}}\right].$$
(5)

The mixed layer is subdivided into *N* sub-layers, each bounded by the altitudes  $h_n$  and  $h_{n+1}$ ( $h_n < h_{n+1}$ ). Assuming no trace gases and optical homogeneity of each sublayer, the optical depth ( $\Delta \tau^{(n)}$ ), single scattering albedo (SSA,  $\omega_0^{(n)}$ ) and phase matrix ( $\mathbf{P}^{(n)}$ ) of the *n*<sup>th</sup> sublayer are contributed by aerosol and Rayleigh molecules only, therefore

$$\Delta \tau^{(n)} = \Delta \tau_{a}^{(n)} + \Delta \tau_{R}^{(n)}, \qquad (6)$$

7 
$$\omega_{0}^{(n)} = \frac{\Delta \tau_{R}^{(n)} + \omega_{0,a}^{(n)} \Delta \tau_{a}^{(n)}}{\Delta \tau_{R}^{(n)} + \Delta \tau_{a}^{(n)}},$$
 (7)

8 and

9 
$$\mathbf{P}^{(n)}(\Theta) = \frac{\Delta \tau_{\mathrm{R}}^{(n)} \mathbf{P}_{\mathrm{R}}^{(n)}(\Theta) + \omega_{0,\mathrm{a}}^{(n)} \Delta \tau_{\mathrm{a}}^{(n)} \mathbf{P}_{\mathrm{a}}^{(n)}(\Theta)}{\Delta \tau_{\mathrm{R}}^{(n)} + \omega_{0,\mathrm{a}}^{(n)} \Delta \tau_{\mathrm{a}}^{(n)}}, \qquad (8)$$

10 where  $\mathbf{P}_{R}$  and  $\mathbf{P}_{a}$  are the Rayleigh and aerosol scattering matrix, respectively; the SSA of aerosol 11  $\omega_{0,a}$  is a function of scattering coefficient ( $K_{sca,a}$ ) and extinction coefficient ( $K_{ext,a}$ ):  $\omega_{0,a} =$ 12  $K_{sca,a}/K_{ext,a}$ ;  $\Delta \tau_{a}^{(n)}$  is the AOD in the  $n^{th}$  sublayer and can be evaluated analytically after 13 considering the aerosol distribution profile (Eq. (1)) according to:

14 
$$\Delta \tau_{a}^{(n)} = \tau_{a} \left[ \operatorname{erf}\left(\frac{h^{(n+1)} - h_{a}}{\sigma_{a}}\right) - \operatorname{erf}\left(\frac{h^{(n)} - h_{a}}{\sigma_{a}}\right) \right] \left[ \operatorname{erf}\left(\frac{h_{\max} - h_{a}}{\sigma_{a}}\right) - \operatorname{erf}\left(\frac{h_{\min} - h_{a}}{\sigma_{a}}\right) \right]^{-1}.$$
(9)

15  $\Delta \tau_{R}^{(n)}$  in Eqs. (6-8) is the Rayleigh optical depth of the  $n^{th}$  sublayer and is evaluated assuming the 16 US standard atmosphere profile (Tomasi et al., 2005; Bodhaine et al., 2007).

17 As functions of aerosol refractive index, shape and size distribution, the elements of  $\mathbf{P}_{a}$  and 18 the quantities  $K_{ext,a}$  and  $K_{sca,a}$  are computed using Mie theory for spherical particles (van de Hulst, 19 1981) and using T-matrix and geometrical optics methods for nonspherical (spheroidal) particles