# Peer review of "(untitled)"

_Atmospheric Measurement Techniques, 2015_

## Referee Comment (RC1) · Anonymous Referee #1 · 11 May 2016

In conclusion, this work is worth for publication in AMT. But I strongly recommend that you resubmit a revised version, because the present form of your manuscript looks like a short dissertation. The manuscript should be recompiled by focusing upon what are new and different against the previous studies.

For an example, you can simplify the section 2. It seems to be just description of RT model, and then the manuscript becomes to be redundancy. You miss the solid curve (extended adding-doubling) in Fig.2. Other figures also should be more effectively treated. Figs.4, 5, 6, 7, 8 and 11 seems to be bored at least for me. I hope you come up with various ideas to make your figures.

Comparison with GRASP is introduced in Abstract, but I found it just in Fig. 12(a).

[Figure]

And even in this figure, the results with GRASP seem to be not so necessary because AERONET data are available there. If you have some more results compared with GRASP, please let me know them.

Anyway I am very interested in this work. I wish the manuscript should be carefully revised based on the scientific concept.

––––––––––––––––––––––––––––––

---

## Referee Comment (RC2) · Anonymous Referee #2 · 16 May 2016

The paper presents a joint retrieval method for aerosol properties and water-leaving radiance from simulated and real AirMSPI observations. The topic is very interesting and suitable for the journal. However, I think the paper needs improvement mainly due to the presentation, the current paper contains too much details. I would expect that the paper can be much shorter and informative if well-organized. As the authors presented, there are mainly three important aspects highlighted in this paper, they are: (1) A bio-optical model was introduced; (2) Spatial smoothness of aerosol and surface properties were performed (3) A new radiative transfer strategy based on the previous work from the authors, thus those three important aspects should be detailed presented and analyzed. Thus I would like to suggest a major revision. Major comments are

listed below: (1) The introduction part need to summarize the current status of aerosol remote sensing over ocean with/without polarization (2) A small introduction about the AirMSPI instrument will help reader to understand the sensitivity study and the retrieval results. (3) Section 2.1, 2.3, 2.4 and 2.7 can be shifted to appendix part, the authors should merge and re-organize the radiative transfer strategy and surface-atmospheric coupling sections. Section 2.6 can be merged into section 2.5 and some more details should be included in the sensitivity study part. (4) Section 3.4 should be summarized at the beginning of this section, like a summary of all assumptions of this algorithm. And a comprehensive sensitivity study is necessary for all the important assumptions here. (5) Section 3 should be re-organized as Section 2. (6) The difference (not absolute values) between Extended adding-doubling and SOS is preferable in Fig. 2 (7) P32, Line 13 - 17, P34 Line 13 – 19, this part illustrates the potential coupling effect between retrieved AOD and SSA, some more detailed analysis, like for instance, a sensitivity study for typical cases is necessary (8) P33, Line 4 – 15, all the analysis is for case of AOD larger than 0.3? Please note that the global mean (both land and ocean) AOD is about 0.25, AOD over ocean can be much lower. (9) A RGB map from AirMSPI can be helpful to understand Fig. 12 (10) Section 4.2, comparison between AirMSPI and other instruments like MISR, POLDER will be interesting.

---

## Author Comment (AC1) · 27 May 2016

We thank the reviewers for their professional comments and suggestions. Detailed responses to Anonymous Referee #1 are provided below:

1. In conclusion, this work is worth for publication in AMT.

Reply: Thank you.

2. But I strongly recommend that you resubmit a revised version, because the present form of your manuscript looks like a short dissertation. The manuscript should be recompiled by focusing upon what are new and different against the previous studies.

Reply: The length of the paper arises from the introduction of three new concepts as summarized by Reviewer 2: (i) An efficient RT model which couples Markov chain and doubling methods; (ii) A retrieval algorithm which obtains water-leaving radiance and aerosol simultaneously by using an empirically adjusted bio-optical model; (iii) Validation of the retrieval algorithm by truth-in-truth-out tests; (iv) Application of the retrieval to analyze real instrument measurements. All these developments are new and differentiate our work from previous studies. However, revision has been made to make the main body better organized (cf. replies below).

3. For an example, you can simplify the section 2. It seems to be just description of RT model, and then the manuscript becomes to be redundancy.

Reply: Section 2 was simplified in the revised paper. Please refer to our reply to the 3rd Comment of Reviewer 2.

4. You miss the solid curve (extended adding-doubling) in Fig.2.

Reply: The solid curve for extended adding-doubling computation in the left two panels of Fig. 2 are present, and they have very good agreement with SOS computations, which are shown as dots. To avoid confusion, we now use different plot style in the right two panels (which show the difference).

5. Other figures also should be more effectively treated. Figs.4, 5, 6, 7, 8 and 11 seems to be bored at least for me. I hope you come up with various ideas to make your figures.

Reply: We believe these are valuable contributions to the ocean color remote sensing community who is interested in knowing whether multi-angular, multi-spectral, and polarized measurements can meet the PACE requirements, and to those aerosol scientists who hope to see the whether the aerosol retrieval accuracy from the proposed algorithm meets the climate requirements. Our view is that the figures are informative as presented.

[Figure]

6. Comparison with GRASP is introduced in Abstract a  And even in this figure, the results with GRASP seem to be not so necessary because AERONET data are available there. If you have some more results compared with GRASP, please let me know them.

Reply: The comparison to GRASP retrieval adds value to our paper by helping us get a sense about how much difference can be caused by two different algorithms as well as whether the difference to AERONET is also seen from other retrieval code. To avoid misleading, however, in the revised abstract we don't give equal weights to AERONET and GRASP as validation, namely we mention in a separate way that the comparison to GRASP is only for the AERONET USC_SeaPRISM case.

7. Anyway I am very interested in this work. I wish the manuscript should be carefully revised based on the scientific concept.

Reply: Thank you for your comments, which have definitely improved the quality of our paper. The revised paper was in the supplement file.

Please also note the supplement to this comment:
http://www.atmos-meas-tech-discuss.net/amt-2015-394/amt-2015-394-AC1-supplement.pdf
* * *

---

## Author Comment (AC2) · 27 May 2016

We thank the reviewers for their professional comments and suggestions. Detailed responses to Anonymous Referee #2 are provided below:

(1) The introduction part need to summarize the current status of aerosol remote sensing over ocean with/without polarization

Reply: As a brief summary of aerosol retrieval algorithms in use by current-generation satellite imagers over ocean, we added two paragraphs (2nd and 3rd) in the introduction.

(2) A small introduction about the AirMSPI instrument will help reader to understand the sensitivity study and the retrieval results.

Reply: We gave a small introduction of AirMSPI in the first paragraph of Section 4 of the original paper.

(3) Section 2.1, 2.3, 2.4 and 2.7 can be shifted to appendix part, the authors should merge and re-organize the radiative transfer strategy and surface-atmospheric coupling sections. Section 2.6 can be merged into section 2.5 and some more details should be included in the sensitivity study part.

Reply: As suggested, Sections 2.1, 2.3, 2.4 and 2.7 are moved into Appendices A-C. And we re-organized the sections of radiative transfer strategy and surface-atmospheric coupling. Section 2.6 has been merged into section 2.5.

(4) Section 3.4 should be summarized at the beginning of this section, like a summary of all assumptions of this algorithm. And a comprehensive sensitivity study is necessary for all the important assumptions here.

Reply: As suggested, we moved Section 3.4 to the beginning of Section 3 (namely as the 3rd paragraph). Regarding the comprehensive sensitivity study on the decoupling of AOD and SSA, retrieval assumptions, and the benefits of using multi-angle viewing, multi-spectral and polarimetric observations, we are currently performing z-score evaluation, information content analysis, and more truth-in-truth-out retrieval tests. So the work is still in progress. It will be worth of another full-length paper. On the other hand, the main purpose of this paper is to formulate the methodology for joint aerosol and water-leaving radiance retrieval. It forms a basis for one of the sensitivity studies we are heading for.

(5) Section 3 should be re-organized as Section 2.

Reply: We give technical details on the retrieval algorithm design in section 3, which should be interesting to algorithm scientists. But we considered your opinion and added before Section 3.1 a reminder to the readers, "In the next three subsections, we will give some details on the design of multi-patch retrieval algorithm. Readers not interested in it could skip over them."

(6) The difference (not absolute values) between Extended adding-doubling and SOS is preferable in Fig. 2

Reply: The difference was shown in the original paper in the right two panels of Fig. 2. However, we have modified the graphical style to clearly differentiate these plots from the left two panels.

(7) P32, Line 13 - 17, P34 Line 13 – 19, this part illustrates the potential coupling effect between retrieved AOD and SSA, some more detailed analysis, like for instance, a sensitivity study for typical cases is necessary

Reply: Such a study would add to the length of the paper. We believe it deserves a separate paper. This work is in progress.

(8) P33, Line 4 – 15, all the analysis is for case of AOD larger than 0.3? Please note that the global mean (both land and ocean) AOD is about 0.25, AOD over ocean can be much lower.

Reply: Line 4-15 are to explain the influence of aerosol loading in retrieving single scattering albedo. Five aerosol loadings from 0.02 to 1.0 are studied, which does cover the typical global AOD loading mentioned by the reviewer.

(9) A RGB map from AirMSPI can be helpful to understand Fig. 12

Reply: Since the ocean site under study is close to uniform across the image (5 x 5 km2, in the yellow frame of Fig. 12a), it is hard to identify the difference even when we use the AirMSPI RGB color map.

(10) Section 4.2, comparison between AirMSPI and other instruments like MISR, POLDER will be interesting.

Reply: AirMSPI is an airborne instrument while MISR and POLDER are satellite-borne.

The number of bands and viewing angles, spatial resolution, and capabilities of polarimetric measurements are different from each other. While we agree that such a comparison would be interesting, it is beyond the scope of the current paper.

The revised paper was in the supplement file.

Please also note the supplement to this comment:
http://www.atmos-meas-tech-discuss.net/amt-2015-394/amt-2015-394-AC2-supplement.pdf

**Supplement:**

**Joint retrieval of aerosol and water-leaving radiance from multi-spectral, multi-**

**angular and polarimetric measurements over ocean**

Feng Xu [a,*], Oleg Dubovik [b], Peng-Wang Zhai [c], David J. Diner [a], Olga V. Kalashnikova [a], Felix C. Seidel [a],

Pavel Litvinov [b], Andrii Bovchaliuk [b], Michael J. Garay [a], Gerard van Harten [a], and Anthony B. Davis [a]

a): Jet Propulsion Laboratory, California Institute of Technology, Pasadena, California, USA

b): Laboratoire d'Optique Atmosphérique, UMR8518, CNRS/Universite Lille-1, Villeneuve d'Ascq, France c): Department of Physics, University of Maryland, Baltimore County, Baltimore, Maryland, USA

*Corresponding author: Feng.Xu@jpl.nasa.gov

**Abstract**

An optimization approach has been developed for simultaneous retrieval of aerosol properties and normalized water-leaving radiance (nLw) from multi-spectral, multi-angular, and polarimetric observations over ocean. The main features of the method are (1) use of a simplified bio-optical model to estimate nLw followed by an empirical refinement within a specified range to improve its accuracy; (2) improved algorithm convergence and stability by applying constraints on the spatial smoothness of aerosol loading and Chlorophyll-a (Chl-a) concentration across neighboring image patches and spectral constraints on aerosol optical properties and on nLw across relevant bands; and (3) enhanced Jacobian calculation by modeling and storing the radiative transfer (RT) in aerosol/Rayleigh mixed layer, pure Rayleigh scattering layers, and ocean medium separately and then coupling them to calculate the field at the sensor. This approach avoids unnecessary and time-consuming recalculations of RT in unperturbed layers in

Jacobian evaluations. The Markov chain method is used to model RT in the aerosol/Rayleigh mixed layer and the doubling method is used for the uniform layers of the atmosphere-ocean system. Our optimization approach has been tested using radiance and polarization measurements acquired by the Airborne Multiangle SpectroPolarimetric

Imager (AirMSPI) over the AERONET USC_SeaPRISM ocean site (6 February 2013) and near the AERONET La

Jolla site (14 January 2013), which respectively reported relatively high and low aerosol loadings. Validation of the results is achieved through comparisons to AERONET aerosol and ocean color products. For comparison, the

USC_SeaPRISM retrieval is also performed by use of the Generalized Retrieval of Aerosol and Surface Properties algorithm (Dubovik et al., 2011). Uncertainties of aerosol and nLw retrievals due to random and systematic instrument errors are analyzed by truth-in/truth-out tests with three Chl-a concentrations, five aerosol loadings, three different types of aerosols, and nine combinations of solar incidence and viewing geometries.

**Keywords**:

Atmosphere and ocean system, polarized radiative transfer, aerosol retrieval, water-leaving radiance retrieval

© 2015. All rights reserved.

**1.  Introduction**

Aerosols exist in the form of airborne suspensions of tiny particles that scatter and absorb sunlight, leading to significant impacts on Earth's energy and water cycles. Quantifying aerosol influences on climate requires accurate determination of their abundances and optical/microphysical properties, which are highly variable spatially and temporally. Aerosol characterization is also crucial for ocean color remote sensing, as the spectral water-leaving radiances account for only 10-15% of the signal observed at the top of the atmosphere (TOA)

and most of the signal arises from atmospheric scattering. Chlorophyll-a concentration, colored dissolved organic matter (CDOM) and other ocean optical properties retrieved from spectral water-leaving radiance provides a measure of ocean productivity and health of ocean ecosystem.

Small over- or underestimates of the aerosol contribution can bias the determinations of these quantities.

When aerosol is major target of retrieval, water-leaving radiance is often empirically estimated or even neglected in operational algorithms employed by current-generation satellite imagers due to their small contribution to TOA signals. For examples, the MODIS Collection 5

algorithm uses zero water-leaving radiance for all but the 550-nm band, where a value of reflectance 0.005 is assumed ($\rho_{550nm} = 0.005$, cf. Remer et al., 2005, 2006). MISR and POLDER

assume zero water-leaving radiance in the red and NIR bands for aerosol retrieval (Kahn et al.

2010; Deuzé et al. 2000). Then observations are searched within a look-up table (LUT) which contains pre-calculated radiation fields for a limited number of aerosol models. The aerosol model with lowest fitting residue is selected as the solution. Depending on the sensitivity of measurements, different types and combinations of aerosol models have been designed. For example, for retrieving aerosol information from multispectral radiance-only observations, MODIS LUT has 20 combinations of fine and coarse aerosol models (Remer et al., 2005, 2006). For retrieving multi-angle multispectral radiance-only observations, MISR LUT has 74 aerosol mixtures (Kahn et al. 2010). For retrieving multi-angle, multispectral, and polarimetric observations, POLDER LUT consists of 12 aerosol models (Deuzé et al. 2000).

As the main disadvantage of LUT approach, the solutions have to be selected from a finite number of aerosol models which might not be sufficiently representative in the relevant parameter space. New research efforts have been proposed to expand the LUT to cover more aerosol models (e.g. Limbacher and Kahn, 2014). An alternative to LUT approach is optimization-based retrieval. It involves a direct inversion of the measurements within the context of a parametric description of the aerosol and surface characteristics that govern the radiation field observed at the TOA. The optimization-based retrieval is featured by a more compact and continuous representation of the relevant parameter space. A review of modern aerosol retrieval algorithms used by airborne and satellite-borne passive remote sensing instruments has been recently given by Kokhanovsky (2015).

[revised manuscript text omitted]

Without using bio-optical models, some RT models for CAOS consider specular reflection by assuming a flat ocean surface (Jin and Stamnes, 1994; Bulgarelli et al., 1999; Chami et al.

2001; Sommersten et al., 2009; Zhai et al., 2009) for simplicity. Better modeling fidelity and accuracy is then achieved by including sea surface roughness into the RT models (Nakajima and

Tanaka, 1983; Fischer and Grassl, 1984; Masuda and Takashima, 1986; Kattawar and Adams,

1989; Mobley, 1994; Deuzé, 1989; Jin et al., 2006; Spurr, 2006) and including the water-leaving radiance and/or ocean foam reflection based on a Lambertian or a more general bidirectional reflectance distribution model (Koepke, 1984; Lyapustin and Muldashev, 2001; Mobley et al.

2003; Sayer et al., 2010; Sun and Lukashin, 2013; Gatebe et al., 2005). Though empirical parameterization of water-leaving radiance simplifies the radiative transfer, the relationship between water-leaving radiance and inherent optical properties (IOP) of dissolved or suspended ocean constituents is indirect. Such a weakness can be overcome by using bio-optical models to relate IOP directly to water-leaving radiance. The bio-optical model based RT methods make it feasible to perform a one-step retrieval of IOP and aerosol optical properties from TOA

measurements of radiance and polarization (e.g., Hasekamp et al. 2011), which is a complementary retrieval strategy to the prevailing two-step retrieval that obtains nLw from TOA

via atmospheric correction and then determines IOP from nLw (IOCCG, 2006). Various RT

solutions involving the use of bio-optical models have been developed and can be used for this purpose. These include the invariant imbedding method adopted by HydroLight (Mobley, 2008)

and its faster version EcoLight (Mobley, 2011a) for scalar (intensity only) RT, and the adding- doubling method (Chowdhary et al., 2006) and successive-order-of-scattering method (Zhai et al., 2010) for polarized RT in the CAOS.

Joint retrieval of aerosol and nLw properties requires supplementing the forward RT

calculations with a sophisticated and computationally efficient inverse model to disentangle their contributions to TOA radiometry and polarimetry. Motivated by the development of a multi- angle imaging polarimeter at JPL—the Airborne Multiangle SpectroPolarimetric Imager (AirMSPI) (Diner et al., 2013)—this paper describes the development of a coupled aerosol-ocean retrieval methodology. Our method (1) employs a simplified bio-optical model to obtain a reasonable estimate of nLw in the first retrieval step, followed by an empirical refinement in the subsequent step; (2) applies constraints on the spatial smoothness of aerosol and Chl loadings across neighboring image patches and spectral constraints on aerosol optical properties and on nLw across relevant bands to improve the convergence and stability of the algorithm; and (3)

models and stores the RT fields in the aerosol/Rayleigh mixed layer, the pure Rayleigh scattering layers, and the ocean medium separately, and then couples them to obtain the radiative field at the sensor—thereby enhancing the Jacobian evaluations by reusing RT fields in the unperturbed layers. The Markov chain and doubling methods are applied to the mixed and uniform layers, respectively, to gain computational efficiency.

The parameters of our retrieval include spectrally dependent real and imaginary parts of aerosol refractive index, aerosol concentrations of different size components, mean height and width of aerosol distribution, nonspherical particle fraction, wind speed over ocean surface, and normalized water-leaving radiance. As auxiliary product, aerosol phase matrix is obtained from the retrieved refractive index and normalized size distribution. Throughout the paper, we use the definition of "exact" normalized water-leaving radiance (nLw) given by Morel et al. (2002). It is consistent with the definition adopted by Franz et al. (2007) and Zibordi et al. (2009) and is related to the remote sensing reflectance ($R_{rs}$) by $R_{rs} = nLw/F_0$, where $F_0$ is the extraterrestrial solar irradiance.

The paper is organized as follows. In Section 2, we introduce our development of the RT model that integrates the Markov chain and adding-doubling methods for CAOS. The multi-patch retrieval algorithm is described in Section 3. In Section 4, a truth-in/truth-out test is performed to assess the retrieval uncertainties for a variety of synthetic scenarios combined from three types of aerosols, five aerosol loadings, three Chl-a concentrations, three solar incidence angles, four viewing geometries, and two types of measurement noise. To test the algorithm with real data, retrievals applied to AirMSPI observations over the USC_SeaPRISM AERONET site and near the La Jolla AERONET site are compared to the independent AERONET results. A summary is presented in Section 5.

**2. A integrated radiative transfer model for a coupled atmosphere-ocean system**

A five-layer model is established for a CAOS system, which consists (from the bottom up) of the ocean medium, the air-water interface, a pure Rayleigh layer, an aerosol/Rayleigh mixed layer, and a second pure Rayleigh layer (see Fig. 1). All layers are vertically homogeneous except for the "mixed layer", where the aerosol has its own vertical distribution profile different than that of the Rayleigh-scattering molecular atmosphere. Parameterizations of distribution profile, size and single scattering properties of aerosols in the mixed layer are demonstrated in Appendix A.

**2.1 Radiative transfer modeling and Jacobian evaluation strategy**

The five-layer CAOS model allows the use of different RT methods to model radiative transfer in different layers based on their computational strength. As an example, the Markov chain method (Esposito and House, 1978; Xu et al. 2010, 2011, and 2012), which exhibits high computational efficiency for modeling RT in vertically inhomogeneous media (Esposito, 1979), is adopted in this work for the aerosol/Rayleigh mixed layer (see Appendix B for details). The doubling method (Stokes, 1862; van de Hulst, 1963; Hansen, 1971; de Haan et al., 1987; Evans and Stephens, 1991; among others), which exhibits high efficiency for modeling RT in optically homogeneous media (Esposito, 1979) is used for the two pure Rayleigh layers and the ocean medium (assumed to be homogeneous throughout the paper). Appendix C gives an example of using doubling method for modeling RT in the ocean medium. The radiative fields from all layers are then coupled using an adding strategy to obtain the TOA fields.

In addition to enable a combination of the strengths of different RT methods, the strategy of separate RT modeling in the five layers also makes for an efficient optimization-based retrieval. During the iterative optimization process, Jacobians are calculated to represent how the radiation fields vary as a function of the model parameters. When they are evaluated by perturbing a model parameter within one of the layers, the diffuse RT fields for all other layers are unchanged from the values obtained from the forward RT simulation and thus can be reused. For example, calculation of the Jacobians with respect to surface or ocean bio-optical parameters does not require re-computation of RT in the atmospheric layer because it has already been derived from the previous forward model calculation. Similarly, when evaluating Jacobians with respect to the aerosol parameters, it is unnecessary to repeat the RT computation of the Rayleigh layers and in the ocean or at the air-water interface. Because optimization-based retrievals involve Jacobian evaluations for a large number of parameters at all iterative steps, this strategy significantly improves the retrieval efficiency.

2.2   Atmosphere-ocean coupling

For the atmosphere, the diffuse radiative fields for the aerosol/Rayleigh mixed layer and the pure Rayleigh layers are computed by Markov chain and doubling methods, respectively, and then coupled to get the diffuse reflection and transmission matrices for the whole atmosphere (see Appendix B).

For the ocean, the radiative field in the bulk medium is computed by doubling method with the optics of ocean constituents evaluated by use of a simplified bio-optical model (see Appendix

D), then coupled with the reflection and transmission across the air-water interface, and finally corrected to account for Raman scattering (see Appendix C). In addition to the contribution by water-leaving radiance from the simplified bio-optical model, total light leaving ocean surface also includes polarized specular reflection ($\mathbf{R}_{W}$, see Appendix E), a Lambertian term for depolarizing ocean foam reflection and an empirical, Lambertian correction term "$\Delta a_{WL}$" to account for the errors of the single-parameter based bio-optical model for water leaving radiance (i.e., departures from the predetermined functional relationships to Chl-a concentration). Thus, the overall bidirectional ocean surface reflection matrix $\mathbf{R}_{surf}$ is described by

$$\pi\mathbf{R}_{surf} = f_{foam}a_{foam}\mathbf{D}_0 + (1-f_{foam})\mathbf{R}_W + (1-f_{foam})\mathbf{R}_{WL}^{Bio} + (1-f_{foam})\Delta a_{WL}\mathbf{D}_0 , \qquad (1)$$

where $\mathbf{D}_0$ is a zero matrix except $D_{0,11}=1$; $a_{foam}$ is foam albedo; $f_{foam}$ is foam coverage fraction related to wind speed W by $f_{foam} = 2.95e\text{-}6\times W^{3.52}$ (Koepke, 1984); and $\mathbf{R}_{WL}^{Bio}$ is the reflection matrix of the ocean-interface system with Raman scattering correction (see Appendix C). Note that $\mathbf{R}_{WL}^{Bio}$ is a physically-based term in which Chl-a concentration is an adjustable free parameter. The last two terms of Eq. (1) constitute our water-leaving radiance model. With and without assuming $\Delta a_{WL}$ to be 0 the simplified and the empirically adjusted bio-optical models are formulated, respectively. Though the water-leaving radiance model in Eq. (1) has angular dependence, to be consistent with the conventional ocean color products we derive from $\mathbf{R}_{WL}^{Bio}$

and  $\Delta a_{WL}$ in  Eq. (1) the normalized water-leaving radiance by setting the Sun at zenith and viewing angle to be nadir, namely

$$nLw = \frac{F_0}{\pi}\left(\frac{d_0}{d}\right)^2 \left[ R_{WL,11}^{Bio}(\theta_v = 0°; \ \theta_0 = 0°; \ [Chl\_a]) + \Delta a_{WL} \right], \tag{2}$$

where $d_0$ is the Earth-Sun distance at which the solar irradiance $F_0$ is reported, and $d$ is the Earth-

Sun distance at the time of measurement. Note that nLw, $\mathbf{R}_W$, $\mathbf{R}_{surf}$, $\mathbf{R}_{WL}^{Bio}$, $\mathbf{R}_{OS}^{Bio, NR}$, $a_{foam}$, $\Delta a_{WL}$, and $F_0$ in Eqs. (1)-(2) are all spectrally-dependent.

Once the diffuse reflection and transmission matrices of the atmosphere and reflection from ocean system are individually known, their coupling to get RT field for the full CAOS is implemented by using the adding method. Two operators $\mathbf{Q}$ and $\mathbf{S}$ are defined to account for the interaction between the ocean and atmosphere via single and higher orders of reflection, respectively,

$$\mathbf{Q}_1 = \mathbf{R}_{atmos}^* \mathbf{R}_{surf} \tag{3a}$$

$$\mathbf{Q}_n = \mathbf{Q}_1 \mathbf{Q}_{n-1} \tag{3b}$$

$$\mathbf{S} = \sum_{n=1}^{\infty} \mathbf{Q}_n \tag{3c}$$

where $\mathbf{R}_{surf}$ is the diffuse reflection matrix from ocean surface and $\mathbf{R}_{atmos}^*$ is the diffuse reflection matrix from atmosphere with light illumination from bottom of the atmosphere. The matrices for downwelling and upwelling diffuse light at the atmosphere-ocean interface are given by

$$\mathbf{D} = \mathbf{T}_{atmos} + \mathbf{S}\exp(-\frac{\tau_{atmos}}{\mu_0}) + \mathbf{S}\mathbf{T}_{atmos} \tag{3d}$$

and

$$U = R_{surf} \exp(-\frac{\tau_{atmos}}{\mu_0}) + R_{surf} D \tag{3e}$$

respectively. The reflection matrix of the full CAOS is,

$$R_{CAOS} = R_{atmos} \exp(-\frac{\tau_{atmos}}{\mu})U + T^*_{atmos} U . \tag{3f}$$

For simplicity in describing the conceptual scheme, the superscript "m" that denotes Fourier series order was not shown in the above expression. In actuality, the TOA radiation fields are reconstructed from all orders of Fourier terms, namely,

$$BRF_{tot} = \pi \sum_{m=0}^{\infty} (2 - \delta_{0m}) R_{CAOS,11}^{(m)} \cos m\phi , \tag{4a}$$

$$qBRF_{tot} = \pi \sum_{m=0}^{\infty} (2 - \delta_{0m}) R_{CAOS,21}^{(m)} \cos m\phi , \tag{4b}$$

$$uBRF_{tot} = \pi \sum_{m=0}^{\infty} (2 - \delta_{0m}) R_{CAOS,31}^{(m)} \cos m\phi , \tag{4c}$$

$$vBRF_{tot} = \pi \sum_{m=0}^{\infty} (2 - \delta_{0m}) R_{CAOS,41}^{(m)} \cos m\phi , \tag{4d}$$

where the bidirectional reflectance factor $BRF_{tot}$ and $DoLP = \frac{\sqrt{qBRF_{tot}^2 + uBRF_{tot}^2 + vBRF_{tot}^2}}{BRF_{tot}^2}$ are used to fit the observation. Since the Sunlight is unpolarized, other matrix entries (namely $R_{CAOS,}$

$_{ij}$, with $j \geq 2$) are not involved in Stokes vector calculation for the diffuse light from the reflection matrix.

Note that the above formalism for modeling RT in a CAOS assumes a horizontally homogeneous atmosphere above a uniform surface, which is known as the independent pixel/patch approximation (IPA) in RT theory (Cahalan et al., 1994). In reality, however, aerosol properties and surface reflection vary across the pixels/patches. To reduce the IPA errors, the single scattering contribution to the total field evaluated by Eq. (4) is replaced by an exact evaluation of radiance along the line of sight. Moreover, for simplicity of model demonstration, our five-layer model assumes the sensor to be located at the TOA. For real airborne measurements, however, the sensor is located inside the atmosphere. Therefore to improve the modeling accuracy, the radiative field is actually computed at the sensor location. This is realized by adding an extra Rayleigh layer above the sensor altitude (e.g. $h > h_{AirMSPI} = 20$ km in our case)

and then use the $\mathbf{U}$ term in the adding method to compute the diffuse upwelling light reaching the sensor. Moreover, ozone correction is made by $BRF_{tot, corr}(\lambda) = BRF_{tot}(\lambda) \exp[-$

$\tau_{ozone}(\lambda)(1/\mu_0 + f_{ozone}/\mu_v)]$, where $\tau_{ozone}$ is the total ozone optical depth and $f_{ozone}$ is the fraction of ozone above the sensor (in our current study $f_{ozone}$ is assumed to be 20% for $h_{AirMSPI} = 20$ km).

The integrated RT model established in the current section will be used as forward model in retrieval, which is to be introduced in the next section.

**3. Optimization approach for joint aerosol and water-leaving radiance retrieval**

Within the framework of optimization-based retrievals for non-linear problems, various approaches have been proposed to invert passive remote sensing data for aerosol, ocean and surface properties. Ideally, the solution vector $\mathbf{x}$ that contains all relevant parameters characterizing aerosol properties, water-leaving radiance and surface reflection is approached in an iterative way by $\mathbf{x}_{k+1} = \mathbf{x}_k - \Delta\mathbf{x}_k$ with $\mathbf{x}_k$ being the solution after k iterations and $\Delta\mathbf{x}_k$ being the increment being obtained by $\Delta\mathbf{x}_k = (\mathbf{J}_k^T)^{-1}\Delta\mathbf{y}_k$, where $\mathbf{J}_k$ is the Jacobian matrix evaluated with $\mathbf{x}_k$, and $\Delta\mathbf{y}_k$ is the difference between model and measurement ($\Delta\mathbf{y}_k = \mathbf{y}(\mathbf{x}_k) - \mathbf{y}_{meas}$). Unfortunately, the determinant of $\mathbf{J}_k$ is often close to 0 and as a result $\mathbf{J}_k$ is ill-conditioned. Therefore, a stable retrieval that ensures convergence to a physically sensible solution must impose constraints such that $\det[\mathbf{J}_k^T(\mathbf{C}_f)^{-1}\mathbf{J}_k + \gamma_{k,1}\mathbf{W}_{k,1} + \gamma_{k,2}\mathbf{W}_{k,2} + \ldots] > 0$ and $\Delta\mathbf{x}_k = [\mathbf{J}_k^T(\mathbf{C}_f)^{-1}\mathbf{J}_k + \gamma_{k,1}\mathbf{W}_{k,1} + \gamma_{k,2}\mathbf{W}_{k,2} + \ldots]^{-}$

$^{1}\Delta\mathbf{y}_k'$, where $\mathbf{C}_f$ is the covariance matrix of the measured signals, $\mathbf{W}_{k,i}$ denotes the imposed various constraints, $\gamma_k$ is a Lagrange multiplier that assigns a weight to the constraint, and $\Delta\mathbf{y}_k'$

incorporates $\Delta\mathbf{y}_k$ and the relevant *a priori* constraints and Lagrange multipliers. Introduction of various types of constraints and/or an *a priori* estimate of $\mathbf{W}$, and establishment of a means for determinant $\gamma_k$ are key elements of optimization-based algorithms. Different approaches include the Levenberg-Marquardt algorithm (Levenberg, 1944; Marquardt, 1963), the Phillips-Tikhonov-

Twomey algorithm (Phillips, 1962; Tikhonov, 1963; Twomey, 1963, 1975), and the Twomey-

Chahine algorithm (Chahine, 1968), as discussed by Dubovik et al. (2004).

To maximize the use of information provided by different remote sensing instruments on aerosol and surface properties, various algorithms have been applied to inverse radiance and polarimetric signals (Kokhanovsky, 2015; Kokhanovsky et al., 2015). For the particular application of AirMSPI aerosol and water-leaving radiance retrievals, an adaptation of the inversion approach of Dubovik (2004) and Dubovik et al. (2008, 2011) is used. This approach considers inversion as a multi-term Least Square Method fitting. This strategy is convenient for using multiple *a priori* constraints simultaneously. Moreover, as suggested by Dubovik et al.

(2008, 2011), additional constraints on temporal or spatial variability of the retrieved characteristics can be used if the retrieval is performed for a group of observed pixels/patches. In the present application, a smoothness constraint is imposed to constrain spatial variation of aerosol properties and Chl-a concentration over a target area of finite size. While the term

"multi-pixel algorithm" is introduced by Dubovik et al. (2011) for POLDER/PARASOL

retrievals with pixel data of ~6 km x 7 km resolution at nadir, the term "multi-patch algorithm" is used here since the AirMSPI pixel resolution is much finer (10 m x 10 m) and 50 by 50 pixels are merged into a "patch" to reduce IPA errors. Moreover, as an extension of what is meant by multi-spectral and multi-angle, even polarimetric, a "multi-pixel" algorithm can be understood as one based on a forward signal model that can predict how radiances escaping from different pixels are physically coupled, which is tantamount to using 3D RT (cf. Langmore et al. (2013)

for a background-aerosol-and-gas-plume retrieval demonstration). To avoid confusion, we use the terminology "multi-patch" here.

Note that though accurate forwarding RT modeling with multiple aerosol species is possible, the increased number of free parameters challenges the ability to retrieve a globally optimized solution in an efficient way. Therefore, as described in Appendix A, a single aerosol species is assumed to represent an "effective" set of aerosol optical properties, size distribution (which may be multimodal), and vertical profile. Five log-normal size distribution components ($N_{sc} = 5$) are used to represent the aerosol size distribution, with median radii and standard deviations optimally chosen and given in Table 1, and size-independent refractive index are assumed.

Retrieval with more than five size components has also been performed and comparison shows that they both retrieve well aerosol optical properties after being optimally set as log-normally shaped (Dubovik et al., 2006). Since five-component based retrieval is faster it is adopted in the current study. Nevertheless, our retrieval leaves the option open for adopting more than five components as well as for retrieving size-dependent refractive index when extra constraints or sensitivity from observation in some observation cases are available.

In the next three subsections, we will give some details on the design of multi-patch retrieval algorithm for joint aerosol and water-leaving radiance retrieval. Readers not interested in it could skip over them.

## 3.1 Multi-patch retrieval algorithm with smoothness constraints

Imposing smoothness constraints on both the spatial variations of aerosol loading and Chl-a concentration and on spectral variations of aerosol optical properties and nLw leads to the minimization of the following cost function in fitting an $N$-patch image (Dubovik et al., 2011),

$$\begin{aligned}
\mathbf{C}(\mathbf{x}) &= \sum_{i=1}^{N} \boldsymbol{\Psi}(\mathbf{x}_i) + \frac{1}{2}\mathbf{x}^{\mathrm{T}}\boldsymbol{\Omega}_{\text{inter-patch}}\mathbf{x} \\
&= \sum_{i=1}^{N}\left[\boldsymbol{\Psi}_{\mathrm{f}}(\mathbf{x}_i) + \boldsymbol{\Psi}_{\mathrm{s}}(\mathbf{x}_i) + \boldsymbol{\Psi}_{\mathrm{a}}(\mathbf{x}_i)\right] + \frac{1}{2}\mathbf{x}^{\mathrm{T}}\boldsymbol{\Omega}_{\text{inter-patch}}\mathbf{x} \\
&= \frac{1}{2}\sum_{i=1}^{N}\left[\Delta\mathbf{y}_i^{\mathrm{T}}\mathbf{W}_{\mathrm{f},i}^{-1}\Delta\mathbf{y}_i + \gamma_{\mathrm{s}}\mathbf{x}_i^{\mathrm{T}}\boldsymbol{\Omega}_{\mathrm{s},i}\mathbf{x}_i + \gamma_{\mathrm{a}}(\mathbf{x}_i - \mathbf{x}_i^*)^{\mathrm{T}}\mathbf{W}_{\mathrm{a},i}^{-1}(\mathbf{x}_i - \mathbf{x}_i^*)\right] + \frac{1}{2}\mathbf{x}^{\mathrm{T}}\boldsymbol{\Omega}_{\text{inter-patch}}\mathbf{x}
\end{aligned} \qquad , \qquad (5)$$

where $\mathbf{x}_i$ is an iterative solution for the set of parameters being retrieved and $\mathbf{x}_i^*$ is an *a priori*

estimate of the solution corresponding to the $i^{\text{th}}$ patch, $\mathbf{x} = [\mathbf{x}_1, \mathbf{x}_2, \mathbf{x}_3, \ldots \mathbf{x}_{\mathrm{N}}]$; $\boldsymbol{\Psi}_{\mathrm{f}}(\mathbf{x}_i)$, $\boldsymbol{\Psi}_{\mathrm{s}}(\mathbf{x}_i)$ and

$\boldsymbol{\Psi}_{\mathrm{a}}(\mathbf{x}_i)$ correspond to the residues of fitting observations, the spectral smoothness constraints, and the *a priori* estimate, respectively; $\boldsymbol{\Omega}_{\mathrm{s,i}}$ is a smoothness matrix for constraining the spectral variation of aerosol optical properties and water-leaving radiances across the relevant bands; $\mathbf{W}_{\mathrm{f}}$

and $\mathbf{W}_{\mathrm{a}}$ are the weighting matrices for measurements and the *a priori* estimate, respectively; $\gamma$

denotes the relevant Lagrange multipliers; $\Delta\mathbf{y}_i$ is the difference between the model and measurements for the $i^{\text{th}}$ patch $[\Delta\mathbf{y}_i = \mathbf{y}(\mathbf{x}_i) - \mathbf{y}_{\text{meas}}]$; and $\boldsymbol{\Omega}_{\text{inter-patch}}$ is the inter-patch smoothness matrix constructed for the patches along two orthogonal directions (u and v) of the image, namely

$$\boldsymbol{\Omega}_{\text{inter-patch}} = \gamma_{\mathrm{u}}\mathbf{S}^{(m_{\mathrm{u}}),\mathrm{T}}\mathbf{S}^{(m_{\mathrm{u}})} + \gamma_{\mathrm{v}}\mathbf{S}^{(m_{\mathrm{v}}),\mathrm{T}}\mathbf{S}^{(m_{\mathrm{v}})} , \qquad (6)$$

where the derivative matrix $\mathbf{S}^{(m)}$ is constructed from the $m^{\text{th}}$ order difference and $\gamma_{\mathrm{u}}$ and $\gamma_{\mathrm{v}}$ are the

Lagrange multipliers and their values are shown in Table 2 for all retrieval parameters.

The optimal solution is approached in an iterative way so that after $k$ iterations, the solution vector $\mathbf{x}_{i,k+1}$ containing parameters of aerosol and surface properties for the $i^{\text{th}}$ patch is updated as

$$\mathbf{x}_{i,k+1} = \mathbf{x}_{i,k} - t_p \Delta \mathbf{x}_{i,k}, \tag{7}$$

where the multiplier $t_p$ $(0 \le t_p \le 1)$ is introduced to improve the convergence of the nonlinear numerical algorithm (Orega and Reinboldt, 1970). Solving the following normal system constructed for the $N$-patches image at the $k^{\text{th}}$ iteration gives the increment of solution for each patch ($\Delta\mathbf{x}_{i,k}$),

$$\left[ \begin{pmatrix} \mathbf{A}_{1,k} & \mathbf{0} & \dots & \mathbf{0} \\ \mathbf{0} & \mathbf{A}_{2,k} & \dots & \mathbf{0} \\ \dots & \dots & \dots & \dots \\ \mathbf{0} & \mathbf{0} & \dots & \mathbf{A}_{N,k} \end{pmatrix} + \mathbf{\Omega}_{\text{inter-patch}} \right] \begin{pmatrix} \Delta\mathbf{x}_{1,k} \\ \Delta\mathbf{x}_{2,k} \\ \dots \\ \Delta\mathbf{x}_{N,k} \end{pmatrix} = \left[ \begin{pmatrix} \nabla\mathbf{\Psi}(\mathbf{x}_{1,k}) \\ \nabla\mathbf{\Psi}(\mathbf{x}_{2,k}) \\ \dots \\ \nabla\mathbf{\Psi}(\mathbf{x}_{N,k}) \end{pmatrix} + \mathbf{\Omega}_{\text{inter-patch}} \begin{pmatrix} \mathbf{x}_{1,k} \\ \mathbf{x}_{2,k} \\ \dots \\ \mathbf{x}_{N,k} \end{pmatrix} \right], \tag{8}$$

where the Fisher matrix for the $i^{\text{th}}$ patch

$$\mathbf{A}_{i,k} = \mathbf{J}_{i,k}^{\text{T}} \mathbf{W}_{\text{f},i}^{-1} \mathbf{J}_{i,k} + \gamma_{\Delta,i}\mathbf{\Omega}_{\Delta,i} + \gamma_{\text{a},i}\mathbf{W}_{\text{a},i}^{-1}, \tag{9}$$

is a function of Jacobian matrix $\mathbf{J}_{i,k}$ and weighting matrix $\mathbf{W}_{\text{f},i}$, and $\nabla\mathbf{\Psi}(\mathbf{a}_{i,k})$ is the gradient of the minimized quadratic form:

$$\nabla\mathbf{\Psi}(\mathbf{x}_{i,k}) = \mathbf{J}_{i,k}^{\text{T}} \mathbf{W}_{\text{f},i}^{-1}(\mathbf{y}_{i,k} - \mathbf{y}_{i,\text{meas}}) + \gamma_{\text{s},i}\mathbf{\Omega}_{\text{s},i}\mathbf{x}_{i,k} + \gamma_{\text{a},i}\mathbf{W}_{\text{a},i}^{-1}(\mathbf{x}_{i,k} - \mathbf{x}_i^*), \tag{10}$$

where $\mathbf{y}_{\text{meas}}$ contains the measurement data; $\mathbf{y}_k$ contains the modeled radiance and polarization with $\mathbf{x}_k$; $\mathbf{W}_{\text{f}}$ is the weighting matrix defined as the covariance matrix $\mathbf{C}_{\text{f}}$ normalized by its first diagonal element namely $\mathbf{W}_{\text{f}} = (1/\sigma_{\text{sd},1}^2)\mathbf{C}$ (with $\sigma_{\text{sd}}$ being the standard deviation); $\mathbf{W}_{\text{a}}$ is the weighting matrix of the *a priori* estimate $\mathbf{x}^*$; and $\mathbf{\Omega}_{\text{s}}$ is the single-patch based smoothness matrix containing sub-smoothness matrices for all parameters. The Lagrange multipliers $\gamma_{\text{s}}$ reflects the strength of the smoothness constraints.

As listed in Table 2, the parameters of the retrieval include spectrally dependent real ($m_r$)

and imaginary ($m_i$) parts of aerosol refractive index, aerosol concentrations of all size components ($C_{v,i}$), mean height ($h_a$) and half width ($\sigma_a$) of aerosol layer, nonspherical particle fraction ($f_{ns}$), wind speed over ocean (W), Chl-a concentration ([Chl_a]) and $\Delta a_{WL}$ which adjust the nLw values in the second step of the retrieval. These parameters form the solution vector $\mathbf{x}$ =

$\log[m_r(\lambda), m_i(\lambda), C_v(r), h_a, \sigma_a, f_{ns}, W, Chl\_a, a_{WL, Const}(\lambda) + \Delta a_{WL}(\lambda)]^T$, where the natural logarithm is used to ensure non-negativity of the real solution after dynamic positive or negative changes during the iterative optimization process. The term $a_{WL, Const}$ is an offset determined from nLw using [Chl_a] from the first retrieval step to ensure that thee adjustment of nLw in logarithmic space is real. Then $\gamma_s \mathbf{\Omega}_s$ is constructed as a block matrix from diagonal concatenation of the spectral smoothness matrices for real and imaginary parts of refractive index and $\Delta a_\lambda$, namely for all patches,

$\gamma_s \mathbf{\Omega}_s = \mathrm{diag}\{\mathbf{0}, \mathbf{0}, \mathbf{0}, \mathbf{0}, \mathbf{0}, \gamma_{s,m_r}\mathbf{\Omega}_{s,m_r}, \gamma_{s,m_i}\mathbf{\Omega}_{s,m_i}, \mathbf{0}, \mathbf{0}, \mathbf{0}, \mathbf{0}, \mathbf{0}, \mathbf{0}, \gamma_{s,(a_{WL,Const}+\Delta a_{WL})}\mathbf{\Omega}_{s,(a_{WL,Const}+\Delta a_{WL})}\}$,  (11)

where $\mathbf{0}$ represents a zero submatrix for a parameter not being subject to any smoothness constraints; and the Lagrange multipliers $\gamma_s$ are pre-determined and given in Table 2.

In our retrieval test, an *a priori* estimate is assumed unavailable so we set $\mathbf{a}_{i,k} = \mathbf{a}_{i,a}^*$.

Therefore Eq. (10) simplifies to

$$\nabla\mathbf{\Psi}(\mathbf{x}_{i,k}) = \mathbf{J}_{i,k}^T \mathbf{W}_{f,i}^{-1}(\mathbf{y}_{i,k} - \mathbf{y}_{i,\mathrm{meas}}) + \gamma_{s,i}\mathbf{\Omega}_{s,i}\mathbf{x}_{i,k}.$$  (12)

When the spectral and spatial smoothness constraints are turned off (namely setting $\gamma_s = \gamma_u = \gamma_v =$

0), the multi-patch algorithm reduces to the traditional Levenberg-Marquardt algorithm (Levenberg, 1944; Marquardt, 1963), which has been used for retrieval tests with MISR

synthetic radiances (Diner et al., 2011; Xu et al., 2012).

Ideally, the retrieval is deemed successful when the minimization of the cost function is achieved, such that

$$2\sum_{i=1}^{N_{patch}} \boldsymbol{\Psi}(\mathbf{x}_{k,i}) + \mathbf{x}_k \boldsymbol{\Omega}_{\text{inter-patch}} \mathbf{x}_k^{\text{T}} \leq N_{\text{inter-patch}} \varepsilon_f^2 + \sum_{i=1}^{N_{patch}} (N_{f,i} + N_{s,i} + N_{a*,i} - N_{a,i})\varepsilon_f^2 , \tag{13}$$

where $N_{f,i}$, $N_{s,i}$, $N_{a,i}$ and $N_{a*,i}$ are the number of observations, spectral smoothness, number of unknowns, and *a priori* estimates of parameters corresponding to i$^{th}$ patch, respectively; $N_{\text{inter-}}$

$_{\text{patch}}$ is the number of spatial smoothness constraints; and $\varepsilon_f^2$ is the expected variance due to measurement errors. In practice, forward RT modeling error and other un-modeled effects can impede realization of the condition shown in Eq. (13). Therefore, the retrieval is also terminated when the relative difference of fitting residues with solutions from two successive iterations drops below a user-specified threshold value, $\varepsilon_c^2$. Namely,

$$\frac{\left[2\sum_{i=1}^{N_{patch}} \boldsymbol{\Psi}(\mathbf{x}_{k+1,i}) + \mathbf{x}_{k+1} \boldsymbol{\Omega}_{\text{inter-patch}} \mathbf{x}_{k+1}^{\text{T}}\right] - \left[2\sum_{i=1}^{N_{patch}} \boldsymbol{\Psi}(\mathbf{x}_{k,i}) + \mathbf{x}_k \boldsymbol{\Omega}_{\text{inter-patch}} \mathbf{x}_k^{\text{T}}\right]}{2\sum_{i=1}^{N_{pixel}} \boldsymbol{\Psi}(\mathbf{x}_{k,i}) + \mathbf{x}_k \boldsymbol{\Omega}_{\text{inter-patch}} \mathbf{x}_k^{\text{T}}} \leq \varepsilon_c^2. \tag{14}$$

is the second criterion to terminate the optimization.

3.2   Determination of Lagrange multipliers

Following Dubovik and King (2000), the Lagrange multipliers reflecting the strength of smoothness constraints are defined as,

$$\gamma_g = \varepsilon_f^2 / \varepsilon_g^2 \text{ and } \gamma_a = \varepsilon_f^2 / \varepsilon_a^2 , \tag{15}$$

where $\varepsilon_f^2$ , $\varepsilon_a^2$ and $\varepsilon_g^2$ are the first diagonal elements of the covariance matrices corresponding   to the measurements ($\mathbf{C}_f$), to the *a priori* estimates ($\mathbf{C}_a$) and to the smoothness constraints ($\mathbf{C}_g$, with the subscript "g" indicating the spectral smoothness constraint "s" or spatial smoothness constraint "u" or "v"), respectively. To estimate $\varepsilon_g^2$ for a given parameter to be retrieved ($x_j$)

which is a function of t, the most unsmooth known solution $x_j^{ns}(t)$ over the target area is used, namely,

$$\varepsilon_g^2 = \int_{t_{min}}^{t_{max}} \left( \frac{d^m[x_j^{us}(t)]}{d^m t} \right)^2 dt \, , \tag{16}$$

where $t_{min}$ and $t_{max}$ specify the lower and upper bound of $t$. In practical implementation of our algorithm, however, the Lagrange multipliers are modified in the following way:

$$\gamma_g^{Final} = \frac{N_f}{N_g} \frac{\tilde{\varepsilon}_f^2}{\varepsilon_f^2} \gamma_g \text{ and } \gamma_a^{Final} = \frac{N_f}{N_a} \frac{\tilde{\varepsilon}_f^2}{\varepsilon_f^2} \gamma_a \, . \tag{17}$$

There are two differences between $\gamma_{...}^{Final}$ and $\gamma_{...}$ :

1. The multipliers "$N_f/N_g$" and "$N_f/N_a$" are introduced to account for possible redundancy of the measured and *a priori* data. Considering that $\varepsilon_{...}^2$ is a variance of the error in a single measured or estimated *a priori* value, if we have $N$ values of similar kind the total variance increases proportionally to $N$. Introducing this coefficient ensures that when there are several kinds of data, the data with fewer values are given comparable weight as the data type for which there is a greater number of available values.

2. The multiplier $\tilde{\varepsilon}_f^2/\varepsilon_f^2$ is introduced with $\tilde{\varepsilon}_f^2$ estimated as the dynamic fitting residual during iterations:

$$\tilde{\varepsilon}_f^2(\mathbf{x}_k) \approx \frac{2 \sum_{i=1}^{N_{patch}} \Psi(\mathbf{x}_{k,i}) + \mathbf{x}_k \mathbf{\Omega}_{inter\text{-}patch} \mathbf{x}_k^T}{N_{inter\text{-}patch} + \sum_{i=1}^{N_{patch}} (N_{f,i} + N_{s,i} + N_{a*,i} - N_{a,i})} \, . \tag{18}$$

With the multiplier $\tilde{\varepsilon}_f^2 / \varepsilon_f^2$, the fitting residual is used as an estimate of measurement error variance. As a result, during the first few iterations the contribution of the *a priori* term is strongest, and its influence decreases as the retrieval progresses. This is done to ensure mostly monotonic convergence, as in the Levenberg-Marquardt procedure (Levenberg, 1944;

Marquardt, 1963). However, the Levenberg-Marquardt approach does not specify a particular scheme for introducing these terms, rather it relies on the implementer's intuition. Our algorithm requires the fitting errors in the initial iterations to be dominated by model linearization errors as opposed to random measurement errors. Because at each iterative step the full forward model is replaced by its linear approximation, the "errors of linearization" decrease as convergence toward the final solution progresses, and they practically disappear so that $\tilde{\varepsilon}_f^2$ becomes equal to

$\varepsilon_f^2$. As a result of this adjustment of the Lagrange multiplier, the non-linear iteration becomes significantly more monotonic.

## 3.3   Implementation of two-step retrieval

As water-leaving radiance is a small contribution to TOA signals, opening a large number of parameters for its retrieval increases the risk of obtaining solutions at local minima of the fitting metric and a significant slowdown of the retrieval. To improve retrieval efficiency and reliability, we use a two-step retrieval strategy: namely, obtaining a reasonable estimate of water- leaving radiance (i.e., close to the truth) by using a bio-optical model constrained by a single parameter ([Chl-a], which governs the abundance of CDOM and phytoplankton in a prescribed way)   during the first step of the retrieval. This is accomplished by setting $\Delta a_{WL}$ to zero so that only Chl-a concentration (the ocean parameter to which the measurements have the largest information content) is retrieved. Other ocean parameters (e.g., CDOM concentration) are models as dependent on [Chl-a]. In   light   of   the   possibility   that   the   bio-optical   model parameterized by Chl-a concentration only can have inaccuracies (particularly in Case 2 waters), this constraint is relaxed in a subsequent step so that the nLw retrieval is improved by letting the

Chl-a concentration and the $\Delta a_{WL}$ term be optimized simultaneously ($\Delta a_{\lambda,WL}$ is allowed to be negative). To mitigate the propagation of instrumental and atmospheric modeling errors to the water-leaving radiance, the second retrieval step 1) allows the adjustment of the bio-optical model based nLw values only within a confined range (e.g. $-15\% \leq \Delta nLw_{adjust}/nLw_{Bio,\,step-1} \leq$

$+15\%$, with $nLw_{Bio,\,step-1}$ being the nLw from the first retrieval step); and 2) imposes a spectral smoothness constraint on nLw($\lambda$).

**4.  Validation of optimization algorithm**

Technologies to extend the observational capabilities of JPL's Multi-angle Imaging

SpectroRadiometer (MISR, Diner et al. 1998) have been developed over the past decade for the purpose of providing additional observational constraints on aerosol and surface properties.

These have been incorporated into AirMSPI, as described in Diner et al. (2013). AirMSPI is an ultraviolet-visible-near-infrared imager that has been flying aboard the NASA ER-2 high altitude aircraft since October 2010. At the heart of the instrument is an 8-band (355, 380, 445, 470, 555,

660, 865, and 935 nm) pushbroom camera mounted on a gimbal to acquire multi-angle observations over a $\pm 65°$ along-track range. Three of AirMSPI's spectral bands (470, 660, and

865 nm) include measurements of the Q and U Stokes polarization parameters. To validate the retrieval approach, the algorithm was applied to simulated and real AirMSPI data.

4.1  Retrievals with simulated AirMSPI observations

Prior to performing retrievals with actual AirMSPI data, truth-in/truth-out tests with simulated data were conducted to assess the accuracy and stability of our optimization approach.

The simulation generates modeled TOA radiance and polarization fields based on AirMSPI

observations over the USC SeaPRISM AERONET-OC site (118.12°W, 33.56°N) off the coast of

Southern California on 6 February 2013. Images of the targeted area were obtained at 9 viewing angles (0°, ±29°, ±47°, ±59°, and ±65°). At nadir, the imaged area covers 10 km x 11 km swath.

The data are mapped to a 10-m spatial grid. Patches comprised of averages of data within 50

pixel x 50 pixel areas were generated, and a total of 102 patches seen at all angles, corresponding to a 5 km x 5 km area, were used simultaneously in the retrievals to take advantage of the multi- patch retrieval algorithm. Totally 126 signals per patch are measured, which include radiances at

9 angles and 8 spectral bands and Q and U at 9 angles and 3 polarimetric bands. Since we use

DoLP in retrieval and did not model or make use of AirMSPI's water-vapor band at 935 nm, in fact we have 90 signals per patch. Moreover, patch-averaged radiance and degree of linear polarization (DoLP) are used in retrieval. The algorithm tests include 3 steps:

(1) Using the AirMSPI observational characteristics described above, simulated measurements were generated for five different aerosol loadings, three aerosol types, three Chl-a concentrations, and nine combinations of Sun illumination and viewing geometries. The five aerosol loadings correspond to AOD of 0.02, 0.1, 0.3, 0.5, and 1.0 in the AirMSPI green band (555 nm). The three aerosol types include (a) weakly absorbing aerosols from the

MODIS/SeaWiFS LUT (Ahmad et al. 2010) with RH = 85% and fine mode volume fraction =

50%; (b) moderately absorbing particles from the same LUT with RH = 30% and fine mode volume fraction = 80%; and (c) dust aerosols (Sokolik and Toon, 1999). Hygroscopic growth is assumed for the water-soluble and smoke aerosols but is excluded for dust aerosols. The refractive index, size parameters, and vertical profile parameters for these three types of aerosols, and the assumed wind speed, are listed in Table 3. The size distributions of the first two aerosol types were fitted by our five-component aerosol size model. The three Chl-a concentrations used were $0.05, 0.2$, and $1 \text{ mg/m}^3$. A perturbation of $\pm10\%$ was imposed on the water-leaving radiance predicted by the Chl-a-based bio-optical model to simulate modeling errors and to test the validity of the two-step retrieval strategy. The wind speed was assumed to be 4 m/s. The mean height and half width of the aerosol distribution profile were set to 1 km and 0.75 km, respectively.

To cover a wide range of observing geometries, a total of nine scenarios based on the

AirMSPI USC_SeaPRISM viewing geometry is used, as illustrated in Fig. 2: the Sun is placed at the original incidence angle $\theta_0 = 49.1°$ as well as at $25°$ and overhead Sun ($\theta_0 = 0°$). Relative azimuth angles of $\phi \approx 50°, 95°, 140°$ and $176°$ are also modeled. The latter case includes glint.

For the case with overhead Sun, only one azimuth angle is necessary.

(2) Random noise was added to the simulated radiance and DoLP values. This is a commonly adopted measure to test the impact of measurement errors on retrieval algorithm performance (Dubovik et al., 2011; Hasekamp and Landgraf, 2005; 2007). We added a relative measurement uncertainty of $\sigma_I = \pm1\%$ to the radiances and an absolute uncertainty of $\delta_{DoLP} =$

$\pm0.005$ to the DoLP. After a random-error test, an extra $\pm4\%$ systematic error was added to study the influence of calibration bias.

(3) Retrieved aerosol properties and Chl-a concentrations were compared to their known (input truth) values.

a)   Influence of aerosol loading and absorption on nLw retrieval

As an example, we use one of the simulated scenarios of AirMSPI observation over

USC_SeaPRISM AERONET-OC site ($\theta_0 = 25°, \phi \approx 95°$) as input. Figures. 3-6 compare retrieved AOD, SSA, particle size distribution (PSD), and nLw, respectively, to the "true" values used in the simulation. In all figures, the top, middle and bottom rows of the panels correspond to

Chl-a concentrations of 0.05, 0.2, and 1 mg/m$^3$, respectively (with ±10% perturbation on water- leaving radiances in different bands). The left, middle and right panels correspond to weakly absorbing, moderately absorbing, and dust aerosols, respectively.

For all aerosol types, the shapes of AOD, SSA, and nLw, as a function of wavelength and

PSD as a function of particle radius, are similar to their "true" values. Due to the limited contribution of nLw to TOA radiance, the aerosol retrieval accuracy is not significantly affected by the Chl-a concentration within the range modeled here. The retrievals over dust are less accurate than for the weakly and moderately absorbing aerosols, due to the fact that dust aerosols are dominated by coarse mode particles and the extinction is more spectrally neutral, so the information provided by the multi-spectral measurements between 355 and 865 nm is less effective to constrain the aerosol retrieval. As expected, Fig. 6 shows higher retrieved nLw accuracy at low AOD loading ($\tau_{555} \leq 0.1$) due to greater atmospheric transparency and increased fraction of nLw in the TOA signals. When the aerosol species changes from weakly absorbing aerosols (corresponding to the three figures in the left column of Fig. 6) to moderately absorbing aerosols (middle column) and then to dust (right column), the bias in nLw increases. This is because the water-leaving radiance signal becomes weaker with increased atmospheric absorption and retrieval of absorbing aerosol properties is more uncertain than for non-absorbing aerosols, and the errors propagate to the water-leaving radiance. As AOD and SSA errors are the largest for dust aerosols, the normalized water-leaving radiance retrieval error also becomes largest in the presence of dust.

A more comprehensive view of aerosol retrieval errors is displayed in Fig. 7a-d. Though the absolute error of retrieved AOD increases as the aerosol loading increases (see Fig. 7a), the relative error of AOD ($100\times|AOD_{retrieved} - AOD_{true}|/AOD_{true}$) generally decreases as the TOA

radiance carries more aerosol information at higher loading (see Fig. 7b). For the same reason, an inverse relationship between aerosol loading and absolute error in single scattering albedo ($|SSA_{retrieved} - SSA_{true}|$) is observed, as shown in Figs. 7c. To evaluate the retrieval error for size distribution, the effective radius is used and calculated for fine and coarse modes by

$$r_{eff,fine} = \left[ \int_{r_{min}}^{r_{cri}} \frac{dv(r)}{d\ln r} d\ln r \right] \left[ \int_{r_{min}}^{r_{cri}} \frac{1}{r} \frac{dv(r)}{d\ln r} d\ln r \right]^{-1} ,$$
(19)

and

$$r_{eff,coarse} = \left[ \int_{r_{cri}}^{r_{max}} \frac{dv(r)}{d\ln r} d\ln r \right] \left[ \int_{r_{cri}}^{r_{max}} \frac{1}{r} \frac{dv(r)}{d\ln r} d\ln r \right]^{-1} ,$$
(20)

respectively, where the lower size limit $r_{min} = 0.04$ μm and the upper size limit $r_{max} = 15$ μm.

Setting $r_{cri}$ to be 0.75 μm for weakly and moderately absorbing aerosols and $r_{cri}$ to be 0.25 μm for dust aerosols to distinguish fine and coarse modes, an generally inverse relationship between aerosol loading and the relative error in effective radius of fine and coarse mode aerosols ($100 \times |r_{eff, retrieved} - r_{eff, true}|/r_{eff, true}$) is also observed for all types of aerosols, as shown in Figs. 7d. For

$\tau_{555} \geq 0.3$, the maximum retrieval error in AOD is ~2.5%, 2.5%, and 7% for weakly, moderately absorbing aerosols, and dust particles, respectively. The maximum retrieval error for SSA $\omega_{0, 355}$

$_{nm}$ is ~0.005, 0.015 and 0.025 for weakly absorbing, moderately absorbing and dust aerosols, respectively. We find that the maximum error in SSA for the weakly absorbing aerosol appears at red and near-infrared bands (660 and 865 nm) for all aerosol loading cases, suggesting that there is less sensitivity to SSA as the ocean reflectance decreases. For the moderately absorbing aerosols, the maximum error is observed at the two UV bands (355 and 385 nm), indicating higher errors as absorption increases, particularly at low aerosol loading. Moreover, increasing

AOD is found helpful to constrain the SSA retrieval for both weakly and moderately aerosols.

However, for dust aerosols, where SSA spans a larger range as a function of wavelength compared to the weakly and moderately absorbing aerosols, limited improvement on SSA retrieval accuracy is gained by increasing AOD.

Figure 7d shows that for weakly and moderately absorbing aerosols the effective radius for coarse mode aerosols has larger retrieval errors than the fine mode aerosol. We attribute this to the fact that the longest spectral band of AirMSPI used in the retrievals (865 nm) is insufficient to fully constrain the coarse mode aerosol PSD.

In Figs. 7e-f, which correspond to Chl-a concentration to be 0.05, 0.2 and 1.0 mg/m$^3$ (with ±10% perturbation imposed on the water-leaving radiance), the retrieval error of normalized water-leaving radiance ($\Delta nLw = nLw_{retrieved} - nLw_{true}$) is plotted against uncertainty metrics specified by the PACE Science Definition Team (SDT) (Del Castillo et al., 2012), i.e., a relative error of 5% or an absolute error of $0.001 \times F_0/\pi$ (whichever is larger) in the visible, and twice these values in the UV. For weakly and moderately absorbing aerosols, the accuracy of nLw at all visible bands mostly falls within the PACE SDT requirement for all aerosol loadings and Chl-a concentrations. The uncertainty in retrieved nLw in the pair of UV bands, however, falls outside the specified bounds when $\tau_{555} > \sim 0.1$. As the TOA signals in the UV are dominated by Rayleigh scattering, accurate retrieval of water-leaving radiance remains challenging even after the inter-patch smoothness constraints on aerosol variation and spectral smoothness constraints on aerosol optical properties are imposed. For all Chl-a concentrations, errors in nLw are largest for dust aerosols, and fall outside the PACE SDT requirement for $\tau_{555} > \sim 0.1$, even in the visible. These errors can potentially be reduced if an improved bio-optical model can be devised that relates the more accurately determined visible nLw values to the values in the UV.

Figure 7 shows that for all aerosol types, even though the retrieval errors of SSA and AOD at low aerosol loading ($\tau_{555}$ < 0.1) are relatively larger than at high AOD, these errors do not propagate to the retrieval of nLw. This is because in the single scattering regime, the path radiance is dominated by scattering optical depth, which is the product of AOD and SSA. This means AOD and SSA errors counteract each other to some extent (i.e., an overestimated AOD is compensated by an underestimated SSA and vice versa) so that scattering optical depth is less biased, leading to a reduced impact on the retrieval of nLw. However, when AOD increases, the fraction of water-leaving radiance in the TOA signal reduces significantly, and accurate separation of its weak contribution in the multiple scattering regime becomes more difficult. The presence of dust aerosols further complicates the retrievals as the aerosols and CDOM share a similar shape of absorption spectra, namely, increasing absorption at shorter wavelengths (Aurin and Dierssen, 2012; Bergstrom et al., 2007).

b)  Effect of multi-patch versus single-patch retrieval

Taking the case of median loading ($\tau_{555}$ = 0.3) of weakly absorbing aerosols and median Chl- a concentration [Chl_a] = 0.2 mg/m$^3$ as an example, Fig. 8a compares simulated single-patch and multi-patch based retrievals of AOD, SSA, PSD, and Chl-a concentration. The Sun illumination and AirMSPI viewing geometry at the USC_SeaPRISM AERONET-OC site on 6 February 2013

is used. While the single patch-based retrieval leads to spatially highly variable Chl-a concentrations with a mean value of 0.26 mg/m$^3$ (namely 30% retrieval error), the multi-patch algorithm yields a more stable and accurate value of 0.21 mg/m$^3$, which is within 5% of the true value. Correspondingly, the accuracy of the nLw retrieval improves by 0.04, 0.03, and 0.01

mW/cm$^2$-sr-μm at 445, 470, and 555 nm respectively, which is a non-negligible amount compared to the PACE tolerated uncertainty 0.07, 0.06, and 0.05 mW/cm$^2$-sr-μm at these bands; the AOD accuracy at 355, 555, 865 nm improves by 3.4%, 6.0%, and 6.4%, respectively; and the

SSA accuracy improves by 0.008, 0.013 and 0.019. For the single patch-based approach, combinations of aerosol type, amount, and nLw that fit the simulated observation are highly non-unique subjected to local optimum solutions. Through the imposition of inter-patch smoothness constraints on aerosol loading and Chl-a concentration, the multi-patch retrieval yields results that are closer to the truth. As indicated in Fig. 8b, the multi-patch algorithm also shows greater noise resistance in all three quantities (nLw, AOD and SSA) simultaneously. The AOD error in the single-patch retrieval decreases as the level of random noise in intensity increases from 0.5% to 2.0%, due to that fact that the errors mainly propagate into nLw and SSA.

c) Comparison to direct water-leaving radiance retrieval

For the same scene parameters used to compare the single- and multi-patch-based retrievals, Fig. 9 compares a retrieval using the bio-optical model and one in which nLw is modeled using unconstrained Lambertian reflectance factors at each wavelength. Using the bio-optical model reduces the parameter space for the water-leaving radiance from 7 independent spectral values to a single parameter (Chl-a concentration) that establishes the spectral variation of the surface signal. While there is little difference between AOD retrieved with and without the bio-optical model, SSA retrieval accuracy improves by 0.01 and 0.02 at 350 nm and 865 nm, respectively. Moreover, a remarkable gain in nLw accuracy by about 6%, 11%, and 12%, or 0.07, 0.12 and 0.03 mW/cm$^2$-sr-μm in absolute magnitude at 445, 470, 555 nm respectively, is achieved when the bio-optical model is used. Given that that the PACE SDT specification tolerates an uncertainty of ~0.06 mW/cm$^2$-sr-μm in these bands, the accuracy gain from using the bio-optical model is significant.

d) Influence of systematic error

The above truth-in/truth out tests were performed assuming instrumental errors are completely random. Such an assumption, however, is not applicable to radiometric errors and their band-to-band variations, which represent systematic deviations from the true values due to calibration errors. For a satellite instrument such as MISR, the radiometric uncertainty is 4% and the band-to-band variations are about 1.5% (Bruegge et al., 2002). Because the absolute error is larger in magnitude than band-to-band error and represents a systematic bias that applies to all measurements, it can potentially have greater impact on retrieval accuracy than band-to-band errors and random noise. To model its effect, we keep the random noise levels used in the previous analysis and add a ±4% systematic error to the simulated radiance signals. The resulting retrieval errors of AOD, SSA, effective radii of fine and coarse mode aerosol, nLw, and band-to- band ratio are displayed in Fig. 10a-f, respectively.

Comparison of Figs. 7 and 10 shows that systematic errors have a larger impact on retrieval accuracy than random errors, as the latter are suppressed by using a lot of patches for retrieval while the former are not. For AOD and SSA, a negative radiance bias causes larger retrieval errors than a positive bias. Comparison of Figs. 10e and 7e shows that errors in nLw due to an intensity bias increase at all AODs: at low aerosol loading the errors propagate to nLw while at high loading the contribution of nLw to the TOA signal is weak, exacerbating errors. On the other hand, comparison of Fig. 10f to Fig. 7h shows a much smaller effect of systematic errors on $\Delta nLw(\lambda)/nLw(555)$; in other words, the systematic errors mainly propagate to the overall magnitude of $nLw(\lambda)$ curve while the relative spectral shape is affected to a much lesser degree.

4.2   Retrievals with real AirMSPI observations

Following algorithm validation using the truth-in/truth-out tests, we applied the algorithm to actual AirMSPI observations acquired over the USC_SeaPRISM AERONET-OC site and near the AERONET site in La Jolla. The USC_SeaPRISM and La_Jolla scenes were chosen from a larger set of AirMSPI field campaign images to ensure cloud free conditions. The data were processed with the recently upgraded data processing pipeline, which includes vicarious radiometric calibrations and improved polarimetric calibration making use of on-board polarization sources. Nadir intensity and DoLP images from combinations of different spectral bands for these two target areas are shown in Fig. 11a and Fig. 12a. Maps of retrieved AOD and

SSA at 555 nm, nLw at 445 nm and 555 nm spectral bands are displayed in Figs. 11b and 12b.

Selecting the image patch that is closest to the AERONET site, our retrieved AOD, SSA, size distribution, and nLw are compared to the independent AERONET results, as shown in Figs.

11c and 12c. We first discuss results from the USC_SeaPRISM retrievals. The AERONET site reported a relatively high 550-nm AOD of 0.30 and 0.26 at 19:08 UTC and 20:08 UTC, respectively, and our retrieval returns an intermediate value of 0.27 from the AirMSPI data acquired at 19:40 UTC. The differences between the AirMSPI and AERONET AOD and SSA

retrievals are within the AERONET SSA retrieval uncertainties (e.g. 0.015 for $\tau_{440}$ and 0.03 for

$\omega_{0,440}$ at $\tau_{440} > 0.2$, see Table 4 of Dubovik et al. 2000). The Generalized Retrieval of Aerosol and

Surface Properties (GRASP) algorithm by Dubovik et al. (2011, 2014) was also run, and the difference between the GRASP and JPL algorithms is on the order of ~0.025 for AOD and

~0.008 for SSA in all bands.

As illustrated in the bottom right panel of Fig. 11c, the retrieved nLw also compares favorably to AERONET reported values. After interpolating AERONET nLw in logarithmic space to obtain nLw in the AirMSPI bands, the differences are found to be 0.0396, 0.0118,

0.0198, and 0.0077 mW/cm$^2$-sr-μm in the 445, 470, 555, and 660 nm bands, respectively. These differences are within the AERONET-OC uncertainties of 0.0462, 0.0516, 0.0279, and 0.0167

mW/cm$^2$-sr-μm in the four bands, obtained by interpolating combined standard uncertainties in validated nLw at various AERONET-OC sites (Gergely and Zibordi, 2014). Note that the nonspherical particle fraction retrieved using both GRASP and JPL algorithm is negligible and the results are not displayed here.

For the second study site, the AirMSPI target area was about 13 km away from the La Jolla

AERONET station. In spite of the distance, the differences between the AirMSPI and

AERONET AOD and SSA values are both within AERONET's uncertainty, as observed from the upper two plots of Fig. 12c. Though the difference in PSD in some size bins falls outside the

AERONET uncertainty range, the bimodality of the size distribution is identified even at the low aerosol loading for this case ($\tau_{555} \sim 0.04$). Independent surface measurements to validate the nLw retrieval were not available at this site.

**5.   Summary and outlook**

Accurate retrieval of both aerosol properties and water-leaving radiance is challenging as the latter only accounts for a small fraction of TOA signals and can be easily contaminated by

Rayleigh and/or aerosol scattering. To ensure high-quality retrievals of the aerosol properties, traditional atmospheric correction schemes, which are focused primarily on retrieval of surface characteristics, may not be sufficient. In light of the additional information provided by multi- angular, multi-spectral, and polarimetric measurements, we tested the concept of simultaneous aerosol and water-leaving radiance retrieval which include spectrally dependent real and imaginary parts of aerosol refractive index, aerosol concentrations of different size components, mean height and width of aerosol distribution, nonspherical particle fraction, wind speed over ocean surface, and normalized water-leaving radiance. An efficient RT modeling strategy has been developed that couples separate runs for modeling RT in two Rayleigh layers, an aerosol/Rayleigh mixed layer, and an ocean medium. Repeated, time-consuming RT

computations for layers whose properties are not perturbed during Jacobian evaluations are avoided. The Markov chain method is used for modeling RT in the mixed layer and the doubling method is used to model RT in the pure Rayleigh layer and ocean medium. These features are implemented to enhance computational efficiency.

Next, an optimization approach has been developed for joint aerosol and water-leaving radiance retrieval. The algorithm involves a two-step retrieval strategy, first relying on a bio-optical model to retrieve a single parameter (Chl-a concentration) that governs nLw, and then allows adjustment of nLw to account for modeling errors. Our optimization algorithm imposes smoothness constraints on the spatial variation of aerosol loading and Chl-a concentration and the spectral variation of aerosol optical properties and nLw. We demonstrated that the use of multi-patch constraints in conjunction with the bio-optical model improves the retrieval accuracy of aerosol properties and water-leaving radiance and stabilizes the algorithm. Truth-in/truth-out tests assuming random errors 1.0% and 0.005 for intensity and DoLP respectively show that the retrieval accuracy of nLw in the visible bands meet the requirements of the PACE SDT in the presence of weakly and moderately absorbing aerosols of optical depth at 555 nm less than 1 and Chl-a concentrations 0.05, 0.2 and 1 mg/m$^3$, whereas meeting the PACE SDT goals in the UV and for dust is more challenging. Increased aerosol absorption reduces the nLw retrieval accuracy except when AOD is low. The addition of systematic errors leads to biases in the absolute magnitude of nLw at both low and high AOD. Band ratios between visible bands (e.g., nLw($\lambda$)/nLw(555)), which are widely used in ocean color analyses, are less impacted by systematic errors for weakly absorbing aerosols. Case studies of AOD, SSA, size distribution and nLw using real AirMSPI observations over the AERONET USC_SeaPRISM OC site and near the AERONET La Jolla site compare favorably to AERONET's reported values.

In future work, the influence of modeling errors on nLw retrievals will be investigated. Since water-leaving radiance accounts for a small fraction of the TOA signals, small forward modeling errors can translate into large nLw retrieval errors. The modeling error can arise from various sources, e.g. neglect of cirrus cloud contamination, approximate treatment of trace-gas absorption and the atmosphere profile, salinity of sea-water, assumption of plane-parallel atmosphere, retrieval of effective aerosol optical properties from assuming single aerosol species and size-independent refractive index, $\delta$-truncation of phase matrix, finite stream number and truncated Fourier terms adopted in the RT model, or errors in the solar spectrum. Further considering the potential errors in our empirically adjusted bio-optical model for optically complex waters (e.g. coastal shallow water and inland water), the combined effects on nLw accuracy remain to be studied. Development of a fast yet accurate CAOS RT model and algorithm validation using a wider set of AirMSPI scenes are also part of our ongoing effort.

**Acknowledgments**

The authors are grateful to Dr. Zia Ahmad at NASA Goddard Space Flight Center for providing the information on aerosol models used in MODIS ocean color retrieval and Dr. Jianwei Wei at Optical Oceanography Laboratory of University of Massachusetts Boston for discussing the AERONET Ocean Color product of normalized water-leaving radiance. This work was performed at the Jet Propulsion Laboratory, California Institute of Technology under contract with the National Aeronautics and Space Administration.

**Appendix A – Parameterizations of distribution profile, size and single scattering**

**properties of aerosols**

The aerosol/Rayleigh mixed layer is defined to have the minimum altitude $h_{min}$ and maximum altitude $h_{max}$. A single aerosol species is assumed to be distributed throughout it with a Gaussian distribution profile characterized by mean height $h_a$ and standard deviation $\sigma_a$ characterizing the width of the aerosol layer. Then, the aerosol concentration profile $c_a$ is

$$c_a(h) = F_{norm} \exp\left[-\frac{(h - h_a)^2}{\sigma_a^2}\right], \tag{A1}$$

where the normalization factor $F_{norm}$ is used to ensure that $\int_{h_{min}}^{h_{max}} c_k(h)\,dh = 1$ and evaluates to

$$F_{norm} = \frac{\sqrt{\pi}\sigma_a}{2}\left[\text{erf}\left(\frac{h_{max} - h_a}{\sigma_a}\right) - \text{erf}\left(\frac{h_{min} - h_a}{\sigma_a}\right)\right], \tag{A2}$$

where erf(x) is the error function.

Breaking the aerosol volumetric size distribution dV(r)/dln(r) into a finite number of size components (Dubovik et al., 2011), the total AOD ($\tau_a$) is the sum of all size components:

$$\tau_a = \sum_{i=1}^{N_{sc}} C_{v,i} K_{ext,a,i} = C_{v,tot} \sum_{i=1}^{N_{sc}} f_i\, K_{ext,a,i}, \tag{A3}$$

where $N_{sc}$ is the total number of size components; $K_{ext,a,i}$ and $C_{v,i}$ are the extinction coefficient (in units of km⁻¹) and column volume concentration (in units of km) of the $i^{th}$ aerosol size component, respectively; $C_{v,tot}$ is the total volume concentration ($C_{v,tot} = C_{v,1} + C_{v,2} + C_{v,3} + \ldots$); and $f_i$ is the volume fraction of the $i^{th}$ component ($f_i = C_{v,i}/C_{v,tot}$).

Moreover, the total aerosol size distribution is constituted as

$$\frac{dV(r)}{d\ln r} = \sum_{i=1}^{N_{sc}} \frac{dV_i(r)}{d\ln r} = \sum_{i=1}^{N_{sc}} C_{v,i} \frac{dv_i(r)}{d\ln r}. \tag{A4}$$

and the associated normalized size distribution is

$$\frac{\mathrm{d}v(r)}{\mathrm{d}\ln r} = \sum_{i=1}^{N_{sc}} f_i \, \frac{\mathrm{d}v_i(r)}{\mathrm{d}\ln r} . \tag{A5}$$

Using a log-normal volume weighted size distribution for all size components, $\mathrm{d}v_i(r)/\mathrm{d}\ln r$ is dimensionless and is parameterized by a median radius for volume size distribution $r_{m,i}$ and a geometric standard deviation $\sigma_i$, namely,

$$\frac{\mathrm{d}v_i(r)}{\mathrm{d}\ln r} = \frac{1}{\sqrt{2\pi}\sigma_i} \exp\left[ -\frac{(\ln r - \ln r_{m,i})^2}{2\sigma_i^2} \right] . \tag{A6}$$

The mixed layer is subdivided into $N$ sub-layers, each bounded by the altitudes $h_n$ and $h_{n+1}$

($h_n < h_{n+1}$). Assuming no trace gases and optical homogeneity of each sublayer, the optical depth ($\Delta\tau^{(n)}$), single scattering albedo (SSA, $\omega_0^{(n)}$) and phase matrix ($\mathbf{P}^{(n)}$) of the $n^{\text{th}}$ sublayer are contributed by aerosol and Rayleigh molecules only, therefore

$$\Delta\tau^{(n)} = \Delta\tau_a^{(n)} + \Delta\tau_R^{(n)} , \tag{A7}$$

$$\omega_0^{(n)} = \frac{\Delta\tau_R^{(n)} + \omega_{0,a}^{(n)}\Delta\tau_a^{(n)}}{\Delta\tau_R^{(n)} + \Delta\tau_a^{(n)}} , \tag{A8}$$

and

$$\mathbf{P}^{(n)}(\Theta) = \frac{\Delta\tau_R^{(n)}\mathbf{P}_R^{(n)}(\Theta) + \omega_{0,a}^{(n)}\Delta\tau_a^{(n)}\mathbf{P}_a^{(n)}(\Theta)}{\Delta\tau_R^{(n)} + \omega_{0,a}^{(n)}\Delta\tau_a^{(n)}} , \tag{A-9}$$

where $\mathbf{P}_R$ and $\mathbf{P}_a$ are the Rayleigh and aerosol phase matrix, respectively; the SSA of aerosol $\omega_{0,a}$

is a function of scattering coefficient ($K_{sca,a}$) and extinction coefficient ($K_{ext,a}$): $\omega_{0,a} = K_{sca,a}/K_{ext,a}$;

$\Delta\tau_a^{(n)}$ is the AOD in the $n^{\text{th}}$ sublayer and can be evaluated analytically after considering the aerosol distribution profile (Eq. (A1)) according to:

$$\Delta\tau_a^{(n)} = \tau_a \left[ \mathrm{erf}\left(\frac{h^{(n+1)} - h_a}{\sigma_a}\right) - \mathrm{erf}\left(\frac{h^{(n)} - h_a}{\sigma_a}\right) \right] \left[ \mathrm{erf}\left(\frac{h_{max} - h_a}{\sigma_a}\right) - \mathrm{erf}\left(\frac{h_{min} - h_a}{\sigma_a}\right) \right]^{-1}. \qquad \text{(A10)}$$

$\Delta\tau_R^{(n)}$ in Eqs. (A7-A9) is the Rayleigh optical depth of the $n^{\text{th}}$ sublayer and is evaluated assuming the US standard atmosphere profile (Tomasi et al., 2005; Bodhaine et al., 2007).

As functions of aerosol refractive index, shape and size distribution, the elements of $\mathbf{P}_a$ and the quantities $K_{ext,a}$ and $K_{sca,a}$ are computed using Mie theory for spherical particles (van de Hulst,

1981) and using T-matrix and geometrical optics methods for nonspherical (spheroidal) particles (Dubovik and King, 2000; Dubovik et al., 2006). During the optimization process, the spectrally dependent refractive index ($m_r + m_i i$) and concentrations ($C_{v,i}$) of the aerosol size components are updated dynamically. To avoid inefficient on-the-fly Mie computations, these particle properties are pre-calculated for all size components and saved on a grid of discrete real and imaginary refractive indices. For an arbitrary combination of real and image refractive indices, interpolation is used to obtain the optical properties. Then, the aerosol phase matrix and scattering and extinction coefficients are updated via linear combination of the contribution of all size components, namely

$$\mathbf{X}_{a,\,ext/sca} = \sum_{i=1}^{N_{sc}} f_i\, \mathbf{X}_{a,\,ext/sca,\,i}, \qquad \text{(A-11)}$$

where $\mathbf{X}$ represents any Mie property of $\{\mathbf{P}_a, K_{ext,a}$ and $K_{sca,a}\}$. Via Eqs. (A7-A9), $\mathbf{X}$ is then mixed with Rayleigh scattering to obtain the overall scattering properties of each layer, which are used as inputs to the RT model for the mixed layer.

**Appendix B – Modeling radiative transfer in atmosphere system**

B.1 Markov chain method for RT in aerosol/Rayleigh mixed layer

The light propagation direction in the mixed layer is discretized into a finite number of angles over the range $0 \leq \mu \leq 1$, where $\mu = |u| = |\cos\theta|$, and $\theta$ is the angle of propagation relative to the downward normal. Within the framework of the Markov chain method, the probability of a photon to transition from one state $(n, u_i)$ to another $(n', u_j)$ is given by the transition matrices

$\mathbf{T}_{\mathrm{Refl}}$ and $\mathbf{T}_{\mathrm{Trans}}$ for diffusely reflected and transmitted light, respectively. The transition probability from state $(n', u_j)$ to emergence from the top and bottom of the mixed layer in direction $u_e$ is given by the extinction matrices $\mathbf{E}_{\mathrm{Refl}}$ and $\mathbf{E}_{\mathrm{Trans}}$, respectively. Given the initial distribution of photons in all states ($\mathbf{\Pi}_0$) from the single scattering computations, the multiple scattering (indicated by subscript "M") contributions to the reflection and transmission matrices of the whole aerosol/Rayleigh mixed layer ("AR") are expressed as a sequence of matrix operations for each azimuthal component $m$ (Xu et al., 2010):

$$
\begin{cases}
(2-\delta_{0m})\mathbf{R}_{\mathrm{M,AR}}^{(m)} = \mathbf{E}_{\mathrm{Refl}}^{(m)}[\mathbf{I}_{\mathrm{d}} - \mathbf{T}_{\mathrm{Refl}}^{(m)}]^{-1}\mathbf{\Pi}_0^{(m)} \\[2ex]
(2-\delta_{0m})\mathbf{T}_{\mathrm{M,AR}}^{(m)} = \mathbf{E}_{\mathrm{Trans}}^{(m)}[\mathbf{I}_{\mathrm{d}} - \mathbf{T}_{\mathrm{Trans}}^{(m)}]^{-1}\mathbf{\Pi}_0^{(m)}
\end{cases},
\tag{B1}
$$

where $\delta_{0m}$ is the Kronecker delta, $\mathbf{I}_{\mathrm{d}}$ is the identity matrix, and $\mathbf{R}_{\mathrm{M,AR}}^{(m)}$ and $\mathbf{T}_{\mathrm{M,AR}}^{(m)}$ are the $m^{\mathrm{th}}$

Fourier sine and cosine components of the mixed layer reflection and transmission matrices, respectively,  namely  $\mathbf{R}_{\mathrm{M,AR}}^{(m)} = [\mathbf{R}_{\mathrm{M,AR,c}}^{(m)}, \mathbf{R}_{\mathrm{M,AR,s}}^{(m)}]^{\mathrm{T}}$  and  $\mathbf{T}_{\mathrm{M,AR}}^{(m)} = [\mathbf{T}_{\mathrm{M,AR,c}}^{(m)}, \mathbf{T}_{\mathrm{M,AR,s}}^{(m)}]^{\mathrm{T}}$.  Analytical expressions for $\mathbf{\Pi}_0^{(m)}$, $\mathbf{E}_{\mathrm{Refl}}^{(m)}$, $\mathbf{E}_{\mathrm{Trans}}^{(m)}$, $\mathbf{R}_{\mathrm{Refl}}^{(m)}$ and $\mathbf{T}_{\mathrm{Refl}}^{(m)}$ have been given by Xu et al. (2010) as a function of optical depth, phase matrix, and SSA for mixed Rayleigh and aerosol scattering (Eqs.

(A7-A9)). Including the contributions of single scattering $\mathbf{R}_{S,AR}^{(m)}$ and $\mathbf{T}_{S,AR}^{(m)}$ gives the total reflection and transmission matrices of the mixed layer, namely

$$\mathbf{R}_{AR}^{(m)} = \mathbf{R}_{M,AR}^{(m)} + \mathbf{R}_{S,AR}^{(m)},$$

(B2)

$$\mathbf{T}_{AR}^{(m)} = \mathbf{T}_{M,AR}^{(m)} + \mathbf{T}_{S,AR}^{(m)}.$$

Equation (B1) is the basic form of the Markov chain method. The majority of computational time is spent in computing the matrix inverse $[\mathbf{I}_d - \mathbf{X}^{(m)}]^{-1}$, with $\mathbf{X}$ being $\mathbf{T}_{Refl}$ or $\mathbf{T}_{Trans}$. To gain computational efficiency, the "chain-to-chain" adding strategy is applied to reduce the matrix dimension via sub-grouping the layers (Esposito, 1979), and a truncated Neumann series expansion is applied to approximate the matrix inverse, namely

$$[\mathbf{I}_d - \mathbf{X}^{(m)}]^{-1} \approx \mathbf{I}_d + \sum_{n=1}^{N_{max}} \prod_{i=1}^{n} \mathbf{X}_i^{(m)}$$

(B3)

Setting 3-4 sublayers for each subgroup, fast convergence and accuracy of matrix inverse computation is usually achieved by using the first 3-4 series terms of Eq. (B3) (namely $N_{max} = 3$

or 4).

B.2 Coupling with doubling method for RT in atmosphere system

The reflection and transmission matrices of the two Rayleigh scattering layers above and below the mixed layer, $(\mathbf{R}_R, \mathbf{T}_R)$, are computed using the doubling method (Hansen, 1971).

Together with the reflection matrix of the mixed layer $(\mathbf{R}_{AR})$ computed from the Markov chain, a set of reflection and transmission matrices $(\mathbf{R}_{atmos}, \mathbf{T}_{atmos})$ for TOA illumination is obtained by applying the adding method twice (e.g. using Eq. (3) of Lacis and Hansen (1974)): two adjacent layers each time. In a similar way, another set of reflection and transmission matrices ($\mathbf{R}_{atmos}^*$,

$\mathbf{T}_{atmos}^*$) corresponding to illuminations from bottom of the atmosphere is evaluated by switching the location of the illumination sources from the top to the bottom of the mixed layer to evaluate ( $\mathbf{R}_{AR}^{*}$ , $\mathbf{T}_{AR}^{*}$ ) and get ( $\mathbf{R}_{R}^{*}$ , $\mathbf{T}_{R}^{*}$ ) from ($\mathbf{R}_{R}$, $\mathbf{T}_{R}$) using the symmetric relationship (Hansen, 1970), and then using the adding method to couple them (e.g. Eq. (4) of Lacis and Hansen (1974)).

**Appendix C – Modeling radiative transfer in ocean system**

In the five-layer CAOS system illustrated in Fig. 1, the ocean system is composed of the ocean medium and the air-water interface. The diffuse reflection matrix of the ocean medium and the reflection and transmission matrices of the air-water interface need to be known before they are coupled to evaluate the diffuse field at the top of ocean.

C.1 Extended adding-doubling method

Evaluation of the reflection matrix of the ocean system follows a similar methodology as for the atmosphere system. However, instead of considering the contributions by molecules and aerosols, RT in the ocean involves scattering and absorption by sea water, CDOM, and phytoplankton and their covariant particles. Evaluation of the IOPs of these components relies on a simplified bio-optical model described in Appendix D, which determines absorption and scattering of CDOM and phytoplankton particles and then bulk optical depth $\tau_{ocean}$, phase matrix

$\mathbf{P}_{ocean}$, and single scattering albedo $\omega_{ocean}$ as a function of Chl-a concentration. We also assume that the ocean components have a uniform vertical distribution, as airborne and satellite-borne passive remote sensing has low sensitivity to the vertical profile. As a consequence of this assumption, the ocean reflection matrix $\mathbf{R}_{ocean}$, which depends on $\tau_{ocean}$, $\omega_{ocean}$, and $\mathbf{P}_{ocean}$, is computed using the doubling method.

As described in Appendix E, reflection of light from ocean surface and its transmission through an air-ocean interface are evaluated using the model of Cox and Munk (1954a; 1954b)

for a wind-roughened ocean surface. The set of reflection and transmission matrices ($\mathbf{R}_W$, $\mathbf{T}_W$)

corresponding to downwelling incident light (in air) and another set of matrices ($\mathbf{R}_W^*$, $\mathbf{T}_W^*$)

corresponding to upwelling incident light (in water) are then determined. In accordance with the adding method, two operators $\mathbf{Q}$ and $\mathbf{S}$ are defined to account for the interaction between the ocean bulk and its interface with air via single and higher orders of reflection, respectively,

$$\mathbf{Q}_1 = \mathbf{R}_W^* \mathbf{R}_{ocean} \qquad \text{(C4a)}$$

$$\mathbf{Q}_n = \mathbf{Q}_1 \mathbf{Q}_{n-1} \qquad \text{(C4b)}$$

$$\mathbf{S} = \sum_{n=1}^{\infty} \mathbf{Q}_n . \qquad \text{(C4c)}$$

However, unlike a real atmospheric layer that attenuates light during its transmission, the air- water interface is a pseudolayer without any thickness, so all attenuation related terms should  be removed. This leads to a modification of the classical adding-doubling scheme (named as the

"extended adding-doubling method" in the remainder of the paper) for coupling the transfer of radiation between the ocean bulk medium and the air-water interface: the matrices describing the downwelling and upwelling of diffuse light at the top of the ocean now become

$$\mathbf{D} = \mathbf{T}_W + \mathbf{S}\mathbf{T}_W \qquad \text{(C4d)}$$

and

$$\mathbf{U} = \mathbf{R}_{Ocean} \mathbf{D} , \qquad \text{(C4e)}$$

respectively, and the reflection matrix describing the upwelling diffusely reflected light leaving the ocean-air interface is

$$\mathbf{R}_{OS}^{Bio, NR} = \mathbf{T}_W^* \mathbf{U} , \qquad \text{(C4f)}$$

where the superscript NR over $\mathbf{R}$ indicates that Raman scattering is not considered at this step (but will be included via a correction introduced in the next section).

As a numerical validation, Fig. C1 compares top-of-ocean radiance and DoLP computed with the extended adding-doubling method via Eq. (C4) and an independent successive-orders- of-scattering code (Zhai et al., 2010). Chl-a concentration was set to 0.30 mg/m$^3$, solar zenith angle to 60°, surface wind speed to 7 m/s and ocean optical thickness to 10. Using 40 streams in the half plane of $0 \leq \mu \leq 1$ and 30 Fourier terms, this case study shows that the maximum relative difference in computed intensity is < 0.3% in magnitude, and the maximum absolute difference in degree of linear polarization (DoLP) is 0.005 in the worst case, and more typically about 0.001.

The difference can be even smaller by using more streams and Fourier terms.

C.2 Correction for Raman scattering

The RT modeling formulated in Section C1 does not account for inelastic scattering processes including Raman scattering by water and fluorescence by chlorophyll and CDOM.

Accurate modeling of these processes is necessary (Mobley, 2008; Zhai et al., 2015) but requires additional inputs and computations that can significantly slow down the retrievals (Mobley,

2011b). To optimize the trade-off between computational efficiency and numerical accuracy, the correction scheme proposed by Lee et al. (2013) is used to quantify the contribution by Raman scattering, namely,

$$
\frac{R_{rs}^{Raman}}{R_{rs}^{NR}} = \varsigma(\lambda) \frac{R_{rs}^{Total}(440)}{R_{rs}^{Total}(550)} + \xi_1(\lambda) \left[ R_{rs}^{Total}(550) \right]^{\xi_2(\lambda)} ,
\tag{C5}
$$

where $R_{rs}^{Total}$ is the total remote sensing reflectance as a sum of Raman scattering ( $R_{rs}^{Raman}$ ) and non-Raman scattering ( $R_{rs}^{NR}$ ); and $\varsigma$ , $\xi_1$ , and $\xi_2$ are model parameters for empirical correction.

Assuming an isotropic distribution of the radiance contributed by Raman scattering, the corrected reflection matrix of ocean and air-water interface system for use by Eq. (1) is,

$$
\mathbf{R}_{WL}^{Bio} = \pi \left[ \mathbf{R}_{OS}^{Bio, NR}(\theta_v, \phi_v; \theta_0) + R_{WL,11}^{Bio,Raman} \mathbf{D}_0 \right].
\tag{C6}
$$

Since the two reference spectral bands at 440 and 550 nm in Eq. (C5) are close to the AirMSPI

bands at 445 and 555 nm, $R_{rs}^{Total}(440)$ and $R_{rs}^{Total}(550)$ are directly replaced by $R_{rs}^{Total}(445)$ and

$R_{rs}^{Total}(555)$ in our calculation. The parameters $\varsigma$, $\xi_1$, and $\xi_2$ for the other AirMSPI bands are obtained by interpolating the values listed for the SeaWiFS bands in Lee et al. (2013).

Fluorescence is neglected in our RT model due to its tiny contribution to TOA signals over open ocean, though it is known to have some impact on nLw at 685 nm (Gordon, 1979).

**Appendix D – A simplified bio-optical model**

As indicated in the last two terms of Eq. (1), our water-leaving radiance model consists of two parts. The first part ($\mathbf{R}_{WL}^{Bio}$) is a physically based term, which is dependent on a single parameter (namely Chl-a concentration, or [Chl_a]). The absorption and scattering properties of colored dissolved organic matter (CDOM or "yellow substance") and phytoplankton and their covariant particles are dependent on this single parameter in a prescribed way. To deal with effects not captured by this model, a second, empirical term ($\Delta a_{WL}$) represented as Lambertian water-leaving radiance adjustment with arbitrary spectral albedo is added. This appendix describes the computation of ocean bulk optical properties as a function of Chl-a concentration, which are then used as input to obtain $\mathbf{R}_{WL}^{Bio,NR}$ via radiative transfer modeling, followed by a Raman scattering correction to compute $\mathbf{R}_{WL}^{Bio}$ (see Appendix C).

Pure sea water, CDOM, and phytoplankton and their covariant particles are considered to be the primary contributors to the oceanic absorption and scattering.

a) Pure sea water

The absorption coefficients of water ($\alpha_w$) are taken from the tabulated experimental data by Pope and Fry (1997). The scattering phase function of pure seawater is (Morel 1974),

$$F_{w,11}(\Theta)=4\pi\times0.06225\times(1+0.835\cos^2\Theta) \tag{D1}$$

where $\Theta$ is scattering angle. To obtain the other entries of the 4 x 4 phase matrix, we use ratios defined by Rayleigh scattering,

$$F_{w,ij}(\Theta)=F_{w,ij}(\Theta) \times F_{R,ij}(\Theta)/F_{R,11}(\Theta), \quad \text{for } i\neq1 \text{ and } j\neq1. \tag{D2}$$

The depolarization factor of sea water is currently set to zero.

Invoking the Einstein-Smoluchowski theory of fluctuation scattering provides $\beta_w$ (Mobley,

1994), and the scattering coefficient for pure seawater is given by

$$\beta_w = 0.00193 \ (550/\lambda)^{4.32}. \tag{D3}$$

Due to the symmetry of scattering function of seawater around 90°, the backscattering coefficient

$\beta_{bw}$ for the sea water is,

$$\beta_{bw} = 1/2 \ \beta_w. \tag{D4}$$

b) Phytoplankton and their covariant particles

Phytoplankton and their covariant particles are assumed to conform to the hyperbolic (Junge) size-distribution, namely,

$$n(r) = \frac{C}{r^{\gamma_p}} \tag{D5}$$

with $n(r)dr$ being the number of particles per unit volume with radius between $r$ and $r + dr$ and $C$

is included to ensure proper normalization after integrating $n(r)$ over all sizes, namely

$$\int_0^\infty n(r)dr = 1. \tag{D6}$$

Knowing the real refractive index of particles ($n_p$) and the slope parameter ($\gamma_p$) of the hyperbolic size distribution, the Fournier-Forland (FF) scattering functionm which is a Mie theory based analytical approximation to the real scattering function of an ensemble of particles, can be determined (Fournier and Forland, 1994; Fournier and Jonasz, 1999), namely,

$$F_{FF}(\Theta) = \frac{1}{(1-\delta)^2 \delta^v} \left\{ v(1-\delta) - (1-\delta^v) + [\delta(1-\delta^v) - v(1-\delta)]\sin^{-2}(\Theta/2) \right\} + \frac{1-\delta_{180}^v}{4(\delta_{180}-1)\delta_{180}^v}(3\cos^2\Theta - 1) \tag{D7}$$

where $\Theta$ is the scattering angle, $\delta_{180}$ is the value of $\delta$ at $\Theta = 180°$, and $\delta$ and $v$ are expressed as

$$v = (3-\gamma_p)/2 \text{ and } \delta = \frac{4}{3(n_p-1)^2}\sin^2(\Theta/2), \tag{D8}$$

respectively. With the FF scattering function, the backscattering efficiency can be obtained analytically (Mobley et al., 2002):

$$B_{bp} = 1 - \frac{1 - \delta_{90}^{v+1} - 0.5(1 - \delta_{90}^{v})}{(1 - \delta_{90}^{v})\delta_{90}^{v}} \tag{D9}$$

where $\delta_{90}$ is $\delta$ evaluated at $\Theta = 90°$.

Mobley (2002) found that the detailed shape of particle scattering function is not critical if a correct backscatter fraction $B_{bp}$ is provided. Characterized by a spectrally neutral backscatter efficiency $B_{bp}$, Huot et al. (2008) obtained an empirical relationship between Chl-a concentration and $B_{bp}$,

$$B_{bp} = \frac{1}{4\pi} \int_{\pi/2}^{\pi} F_p(\Theta)\sin\Theta \, d\Theta = 0.002 + 0.01(0.5 - 0.25\log_{10}[\text{Chl\_a}]) \,. \tag{D10}$$

The spectrally neutral assumption for the backscattering efficiency also indicates that the refractive index and slope parameter are not independent to each other. Knowing $B_{bp}$ from a given Chl-a concentration via Eq. (D10) and further assuming a linear relationship between $n_p$

and $\gamma_p$ (Mobley et al., 2002), namely,

$$n_p = 1.01 + 0.1542(\gamma_p - 3) \,, \tag{D11}$$

Thus, given Chl-a concentration $B_{bp}$ is computed from Eq. (D10). Then Eqs. (D9) and (D11) can be solved to determine $n_p$ and $\gamma_p$ – the two model parameters of the FF scattering function. Figure

D1 illustrates the resulting relationships between $n_p$ and $B_{bp}$, between $\gamma_p$ and $B_{bp}$, and between $n_p$

and $\gamma_p$.

The absorption coefficients of phytoplankton and their covariant particles for $400 \leq \lambda \leq$

nm are parameterized by Bricaud et al (1998) as,

$$\alpha_p = A_p(\lambda)[\text{Chl\_a}]^{E_p(\lambda)} \tag{D12}$$

Integrated with Vasilkov et al (2005)'s $A_p(\lambda)$ and $E_p(\lambda)$ spectra for $300 \leq \lambda \leq 400$ nm from coastal California water measurements, Morrison and Nelson's $A_p(\lambda)$ spectra for $300 \leq \lambda \leq 750$

nm from Bermuda Atlantic Time Series (BATS) site measurements (Morrison and Nelson,

2004), and setting $A_p(\lambda)$ and $E_p(\lambda)$ to 0 beyond 720 nm, the $A_p(\lambda)$ and $E_p(\lambda)$ spectra for $300 \leq$

$\lambda \leq 1000$ nm are available from http://www.oceanopticsbook.info and adopted here.

The particle scattering coefficients are evaluated based on the model by Morel and

Maritorena (2001):

$$\beta_p = \beta_p(\lambda_0) \, (\lambda/\lambda_0)^\kappa \tag{D13}$$

where $\beta_p(\lambda_0)$ is the scattering coefficient at the reference wavelength $\lambda_0$. Following Huot et al.

(2008), we use $\lambda_0 = 660$ and,

$$\beta_p(660) = 0.347 \, [Chl\_a]^{0.766}, \text{ with} \tag{D14}$$

$$\kappa = 0.5(\log_{10}[Chl\_a]-0.3), \quad 0.02<[Chl\_a]<2 \text{ mg/m}^3 \tag{D15}$$

$$\kappa = 0, \qquad [Chl\_a]<0.02 \text{ mg/m}^3. \tag{D16}$$

c) CDOM

Absorption of CDOM ($a_{CDOM}$) is estimated using the model of Bricaud et al. (1981):

$$\alpha_{CDOM}(\lambda) = \alpha_{CDOM}(\lambda_0)\exp[-S(\lambda-\lambda_0)], \tag{D17}$$

where for the reference wavelength $\lambda_0 = 440$ nm, $S = 0.014$ and according to Bricaud et al.

(1998),

$$\alpha_{CDOM}(440) = 0.2[\alpha_w(440) + \alpha_p(440)]. \tag{D18}$$

The scattering coefficient for CDOM is treated as zero in the present study.

d) Total inherent optical properties of sea water

Summarizing the contribution of all components gives the total absorption coefficient of ocean bulk ($\alpha_{\text{ocean}}$, cf. Zhai et al., 2010; Chowdhary et al., 2012):

$$\alpha_{\text{ocean}} = \alpha_w + \alpha_{\text{CDOM}} + \alpha_p, \tag{D19}$$

and the total scattering coefficient:

$$\beta_{\text{ocean}} = \beta_w + \beta_p. \tag{D20}$$

The total scattering function for sea water is

$$P_{\text{ocean},11}(\Theta) = [\beta_w F_w(\Theta) + \beta_p F_{FF}(\Theta)]/\beta_{\text{ocean}} \tag{D21}$$

Polarized radiative transfer computations require the full phase matrix of bulk ocean scattering.

To this purpose, we construct other phase matrix entries ($i \neq 1$ and $j \neq 1$) by using the ratio of measured sea water phase matrix entries, namely,

$$P_{\text{ocean},ij}(\Theta) = P_{\text{ocean},11}(\Theta) \times [F_{VF,ij}(\Theta)/F_{VF,11}(\Theta)], \tag{D22}$$

where the ratio "$F_{VF,ij}(\Theta)/F_{VF,11}(\Theta)$" is taken from the averaged experimental measurements of

Voss and Fry (1984).

Taking the geometric thickness of ocean as $\Delta H$, the total ocean optical thickness is then obtained from

$$\tau_{\text{ocean}} = [\alpha_{\text{CDOM}} + (\alpha_w + \beta_w) + (\alpha_p + \beta_p)]\Delta H = (\alpha_{\text{ocean}} + \beta_{\text{ocean}})\Delta H, \tag{D23}$$

and the ocean single scattering albedo is

$$\omega_{\text{ocean}} = \beta_{\text{ocean}}/(\alpha_{\text{ocean}} + \beta_{\text{ocean}}). \tag{D24}$$

With $\tau_{\text{ocean}}$, $\omega_{\text{ocean}}$ and $\mathbf{P}_{\text{ocean}}$, the reflection matrix of ocean and air-water interface system

$\mathbf{R}_{\lambda,\text{WL}}^{\text{Bio, NR}}$ is determined from radiative transfer modeling (see Appendix C). Further inclusion of a

Raman scattering correction via Eq. (C6) yields $\mathbf{R}_{\lambda,\text{WL}}^{\text{Bio}}$ for the bio-optical model-based water- leaving radiances. As [Chl_a] is the only independent parameter in the simplified model, modeling errors are unavoidable. To account for them, the water-leaving radiances are adjusted in the second retrieval step by allowing $\Delta a_{WL} \neq 0$ in Eq. (1).

**Appendix E – Reflection and transmission matrices of the air-ocean interface**

a) Surface reflection matrix and transmission matrix of the air-ocean interface

With the micro-facet assumption of oceanic wave structure, the polarized ocean surface reflectance is modeled as (Tsang 1985; Mishchenko, 1997),

$$\mathbf{R}_W = \frac{\pi P_S(z_x, z_y) S_h(\cos\theta_v, \cos\theta_i)}{4\cos^4\beta \cos\theta_i \cos\theta_v} \mathbf{r}(\pi - i_2) \mathbf{F}_r(n_w, \theta_i) \mathbf{r}(-i_1),$$
(E1)

where $\mathbf{F}_r$ is the Fresnel matrix for reflection as a function of the refractive index of water ($n_w$)

and incidence angle $\theta_i$; the rotation matrices $\mathbf{r}(\pi-i_2)$ and $\mathbf{r}(i_1)$ are dependent on the angles $i_1$ and $i_2$

which account for the Stokes vector rotations between the meridian and reflection planes (Hovenier,    1969); $\Theta$ is the scattering angle; $\beta$ is the tilt angle of the facet surface normal; $S_h(\mu, \mu_0)$

is a shadowing function (Smith 1967, Sancer 1969, and Brown 1980); and $z_x$ and $z_y$ are the two components of surface slope,

$$z_x = \frac{-\sin\theta_v \sin\phi}{\cos\theta_0 + \cos\theta_v}$$
(E2)

$$z_y = \frac{\sin\theta_0 + \sin\theta_v \sin\phi}{\cos\theta_0 + \cos\theta_v}$$
(E3)

where $\theta_0$ and $\theta_v$ are solar incidence and viewing angles, respectively, and $\phi$ is the relative azimuth angle. Without consideration of the wind direction, the wave slope probability distribution conforms to Cox and Munk's model (1954a; 1954b):

$$P_S(z_x, z_y) = \frac{1}{2\pi\sigma^2} \exp(-\frac{\tan^2\beta}{2\sigma^2}), \text{ with } \tan^2\beta = z_x^2 + z_y^2$$
(E4)

where the slope variance is related to the wind speed (W) by $\sigma^2 = [0.003 + 0.00512W]/2$.

For the downwelling light, the transmission matrix is (Zhai et al., 2010),

$$\mathbf{T}_{\mathrm{W}} = \left[ \frac{n_{\mathrm{w}}^2 \cos\theta_{\mathrm{t}} \cos\theta_{\mathrm{i}}}{(n_{\mathrm{w}} \cos\theta_{\mathrm{t}} - n_{\mathrm{i}} \cos\theta_{\mathrm{i}})^2} \right] \frac{\pi P_{\mathrm{S}}(z_x, z_y) S_{\mathrm{h}}(\cos\theta_{\mathrm{v}}, \cos\theta_{\mathrm{i}})}{4\cos^4\beta \cos\theta_{\mathrm{i}} \cos\theta_{\mathrm{v}}} \mathbf{r}(\pi - i_2) \mathbf{F}_{\mathrm{t}}(n_{\mathrm{w}}, \theta_{\mathrm{i}}) \mathbf{r}(-i_1) , \qquad \text{(E5)}$$

in which, compared to the reflection matrix, the Fresnel matrix for transmission $\mathbf{F}_{\mathrm{t}}$ is used and the extra term in the bracket accounts for the beam convergence when the light transmits from air though the air-water interface.

The equations (E1) and (E5) also apply to the evaluation of reflection and transmission matrices $\mathbf{R}_{\mathrm{W}}^*$ and $\mathbf{T}_{\mathrm{W}}^*$ for the incidence of upwelling light from water into air after substituting

"$n_{\mathrm{w}}$" with the new relative refractive index "$1/n_{\mathrm{w}}$".

**Tables**

Table 1. Median radius ($r_m$) and standard deviation ($\sigma$) of $N_{sc} = 5$ volume weighted log-normal size components, namely $dv_i(r)/d\ln r$ in Eqs. (A4-A5)

| Bin number | Median radius ($r_m$, μm) | Standard deviation ($\sigma$) |
|---|---|---|
| 1 | 0.1 | 0.35 |
| 2 | 0.1732 | 0.35 |
| 3 | 0.3 | 0.35 |
| 4 | 1 | 0.5 |
| 5 | 2.9 | 1 |

Table 2. Parameters in ocean retrieval and Lagrange multipliers for smoothness constraints

| | Range | Order of finite difference for spectral smoothness constraints ($m_s$) | Lagrange regularization factor ($\gamma_s$) | Order of finite difference for inter-patch smoothness constraints ($m_{(u,v)}$) | Lagrange regularization factor $\gamma_{(u,v)}$ |
|---|---|---|---|---|---|
| **Aerosol parameters:** | | | | | |
| Volume concentration of size components ($C_{v,i}$, µm³/µm²) | [1.0E-6, 5] | - | - | 1 | 1 |
| Mean height of aerosol distribution profile ($h_a$, km) | [0.05, 10] | - | - | 1 | 0.01 |
| Width of aerosol distribution profile ($\sigma_a$) | [0.5, 2.5] | - | - | 1 | 0.01 |
| Refr. index (real part: $n_r(\lambda)$) | [1.33, 1.60] | 1 | 0.1 | 1 | 10 |
| Refr. index (imag. part: $n_i(\lambda)$) | [5e-7, 5e-1] | 2 | 0.01 | 1 | 1 |
| Nonspherical particle fraction ($f_{ns}$)† | [1e-3, 1] | - | - | 1 | 0.1 |
| **Surface parameters:** | | | | | |
| Adjustment term ($\Delta a_{WL}(\lambda)$, mW/cm²-sr-µm) | π/F0×(d/d0)²×nLw1× [-15%, +15%]§ | 3‡ | 0.1‡ | 3‡ | 0.1‡ |
| Chlorophyll a concentration ([Chl_a], mg/m³) | Step-1 for [Chl_a]1: [0.02, 15] Step-2 for [Chl_a]2: [0.85, 1.15]× [Chl_a]1 | - | - | 1 | 0.01 |
| Normalized water-leaving radiance^i (nLw (λ), mW/cm²-sr-µm) | - | - | - | - | - |
| Surface wind speed (W, m/sec) | [1, 30] | - | - | 1 | 0.1 |

†: Nonspherical (spheroidal) particle fraction is excluded in truth-in-truth-out test but included in real data retrieval

§: The subscript "1" of "nLw" means the normalized water-leaving radiance determined from Chl-a concentration ([Chl_a]1) retrieved at step-1

‡: Determined with the consideration of constant offset π/F0×(d/d0)²×nLw1(λ)

i: The normalized water-leaving radiance is not directly retrieved. After the second step retrieval, the updated Chl-a concentration [Chl_a]2 and the adjustment term $\Delta a_{WL}$ are used to derive it via Eq. (2)

Table 3. Cases for truth-in/truth-out retrieval tests

| | Weakly absorbing aerosol | Moderately absorbing aerosol | Dust aerosol |
|---|---|---|---|
| **Aerosol** | | | |
| Targeted AOT at 555 nm | 0.02, 0.1, 0.3, 0.5, 1.0 | | |
| Volume fractions ($f_{v,1-5}$) | 4%, 32%, 20%, 4%, 40% | 16%, 56%, 6%, 6%, 16% | 2%, 8%, 1%, 24%, 65% |
| Mean height of aerosol distribution profile ($h_a$, km) | 1 | | |
| Half width of aerosol distribution profile ($\sigma_a$, km) | 0.75 | | |
| Refractive index (mean of real part $n_r(\lambda)$) | 1.388 | 1.522 | 1.497 |
| Refractive index (mean of imag. part: $n_i(\lambda)$) | 1.98E-3 | 1.32E-2 | 3.11E-3 (355 nm) 1.68E-3 (470 nm) 1.03E-4 (865 nm) |
| **Surface** | | | |
| Chlorophyll a ([Chl_a], mg/m$^3$) | 0.05, 0.2, 1.0 | | |
| Adjustment term ($\Delta a_{WL}(\lambda)$, mW/cm$^2$-sr-$\mu$m) | corresponding to $\pm 10\%$ perturbation on bio-optical model simulated nLw at AirMSPI 355, 385, 445, 475, and 550, 660 and 865 nm spectral bands | | |
| Surface wind speed (W, m/sec) | 4 | | |

**Figures**

[Figure]

Fig. C1. Comparison of top-of-ocean radiance (L) and degree of linear polarization (DoLP) computed by the extended adding-doubling model (solid lines in the left two panels) and successive-orders-of-scattering (labeled as

"SOS", dots in the left two panels) with the bio-optical model described in Appendix D for an ocean system (ocean and air-water interface only, no atmosphere). Shown in the top right panel is the percentage difference of reflectance calculated by $100\times(L_{EAD}- L_{SOS})/L_{EAD}$, where the subscript "EAD" denotes our extended adding-doubling method.

Shown in the bottom right panel is the difference of DoLP computed by $100\times(DoLP_{EAD} - DoLP_{SOS})$. The chlorophyll concentration is $[Chl\_a] = 0.30$ mg/m$^3$ and the solar zenith angle is 60°. The ocean surface is roughened by a wind of speed 7 m/s and ocean optical thickness is set to 10. An arbitrary combination of refractive index and slope parameter ($n_p$=1.05, $\gamma_p$= 3.71) is chosen to compute the Fourier-Forland phase function. The results are plotted for viewing angles ($\theta_v$) increasing from 0° to 87° with an angular step of 3°; the five azimuthal planes ($\phi_v$) are 0°, 45°,

90°, 135° and 180° (shown in black, red, blue, green, and cyan, respectively) with respect to the principal plane (namely O-XZ in Fig. 1).

[Figure]

Fig. D1. Left panel: refractive index ($n_p$) of phytoplankton and their covariant particles as a function of backscattering efficiency ($B_{bp}$); Middle panel: slope parameter ($\gamma_p$) as a function of backscattering efficiency ($B_{bp}$); and right panel: refractive index ($n_p$) as a function of the slope parameter ($\gamma_p$). $B_{bp}$ is computed from Eq. (D11) as a function of Chl-a concentration [Chl_a]. The refractive index ($n_p$) and slope parameter ($\gamma_p$) characterizing a Junge size distribution are then determined by solving Eqs. (D10) and (D12) numerically.

[Figure]

Fig. 1. Depiction of the 5-layer CAOS model. A Gaussian vertical distribution profile for aerosols in the mixed layer is assumed and the Markov chain model is used for RT in this optically inhomogeneous layer. The ocean medium and the two Rayleigh layers (below and above the mixed layer, respectively) are treated as optically homogeneous and the doubling method is used for the RT computations. Coupling of these layers and inclusion of the air-water interface are completed by use of the adding strategy. The Sun illuminates the top-of-atmosphere with solar zenith angle $\theta_0$ and azimuthal plane $\phi_0$. We define $\phi = \phi_v - \phi_0$, where the sensor views the atmosphere at viewing angle $\theta_v$ and azimuthal angle $\phi_v$.

[Figure]

Fig. 2. Simulation geometries based on AirMSPI observations over the AERONET OC-site USC_SeaPRISM on 6 February 2013. The three red dots indicate the Sun's location ($\theta_0 = 49.1°$, the actual value at the time of the AirMSPI overflight, as well as 25° and 0°). For each incidence angle, four viewing geometries corresponding to the azimuthal angles $\phi \approx 50°$, 95°, 140°, and 176° are simulated, which are marked in different colors: black, blue, dark red and dark yellow, respectively. Due to symmetry, only one azimuthal plane is necessary to simulate for zenith Sun location. Therefore totally nine geometries are created for truth-in/truth-out test. The viewing angles corresponding to the 9 AirMSPI images form line segments. Each line segment is composed of densely sampled cross-track positions contributed by all patches in the image. For each azimuthal case, a total of nine segments are plotted.

[Figure]

Fig. 3. Simulated true and retrieved spectral AOD for different scene conditions. Left column of three panels: weakly absorbing aerosol; Middle column of three panels: moderately absorbing aerosol; Right column of three panels: dust aerosol. AOD is retrieved for three values of Chl-a concentration: 0.05 (top row of panels), 0.2 (middle row of panels), and 1.0 mg/m$^3$ (bottom row of panels), with ±10% perturbation of water-leaving radiance. Five aerosol loadings, corresponding to $\tau_{555}$ = 0.02, 0.10, 0.30, 0.50, and 1.0, are plotted in dark blue, dark red, dark yellow, purple, and green respectively. The lines with crosses at the AirMSPI wavelengths represent the true AODs, while the open circles correspond to the retrieved values. The synthetic data is from one of the simulated scenarios of AirMSPI observation over USC_SeaPRISM AERONET-OC site ($\theta_0 = 25°$, $\phi \approx 95°$). Though not plotted, the spatial variation of the retrieved AOD across the whole image is less than 1% for all spectral bands.

[Figure]

Fig. 4. Panel layout as in Fig. 3 but for retrieved single scattering albedo. The black line with dots placed at the AirMSPI wavelengths represents the true SSA. The colored symbols represent retrieved SSA for various values of AOD.

[Figure]

Fig. 5. Panel layout as in Fig. 3 but for retrieved normalized aerosol size distribution. The black lines correspond to the true size distribution, with dots at discrete values of particle radius. The colored lines represent retrieved size distributions for various values of AOD.

[Figure]

Fig. 6. Panel layout as in Fig. 3 but for retrieved values of nLw (mW/cm$^2$-sr-um). The black lines correspond to the true nLw, with dots placed at the AirMSPI wavelengths. The colored symbols represent retrieved nLw for various values of AOD.

[Figure]

(a)

(b)

[Figure]

(c)

(d)

[Figure]

                        (e)

                        (f)

[Figure]

(g)

(h)

Fig. 7. Retrieval errors of (a) AOD; (b) AOD (relative difference); (c) SSA; (d) effective radii for fine and coarse mode aerosols; (e)-(g) nLw (signed difference) corresponding to Chl-a concentrations 0.2, 0.05 and 1.0 mg/m$^3$, respectively (with ±10% perturbation imposed on the water-leaving radiance); and (h) band ratios ($R_{\lambda, nLw}$ =

nLw($\lambda$)/nLw(555)). The retrieval errors of aerosol properties show similar features for all Chl-a concentrations.

Therefore the results corresponding to [Chl_a] = 0.2 mg/m$^3$ are displayed in (a)-(d). Via truth-in/truth-out tests, the uncertainties are estimated for AirMSPI multi-spectral, multi-angular, and multi-polarimetric observations over a 5

km x 5 km ocean area. The simulation is based on nine combinations of Sun incidence and viewing geometries.

Relative random noise of 1.0% is used for radiance and absolute random noise of 0.005 is used for DoLP. The colors correspond to seven different AirMSPI spectral bands. The maximum water-leaving radiance error target specified by the PACE Science Definition Team (SDT) is plotted as black curves. The uncertainty of nLw at 865 nm is not displayed since the PACE SDT did not specify a requirement on this band. The spread of the error, depicted by the vertical bars, reflects the dependence on illumination and viewing geometries.

[Figure]

(a)

(b)

Fig. 8. Comparison of single-patch and multi-patch based retrievals of (a) AOD, SSA, aerosol size distribution and Chl-a concentration for the median AOD ($\tau_{555} = 0.3$) of weakly absorbing aerosols and Chl-a concentration of 0.2 mg/m$^3$. The simulation uses the Sun and viewing geometry corresponding to the AirMSPI overflight of the USC_PRISM AERONET site. Image-averaged Chl-a concentrations are 0.29 mg/m$^3$ and 0.22 mg/m$^3$ for the single and multi-patch based retrievals, respectively. A random error of 1.0% and 0.005 is added to the simulated intensity and DoLP data, respectively; (b) AOD, SSA, and nLw (mW/cm$^2$-sr-μm) retrieved with different levels of random noise (0.5%, 1.0%, 2.0% and 3.0%) added to the simulated BRF while the noise in DoLP is kept at 0.005. The aerosol loading, Chl-a concentration, and Sun and viewing geometry are the same as in Fig. 8a.

[Figure]

Fig. 9. AOD, SSA, aerosol size distribution, and nLw retrieved using the bio-optical model compared to retrievals in which water-leaving radiance is modeled simply as Lambertian with arbitrary albedo.

[Figure]

(a)

(b)

[Figure]

(c)

(d)

[Figure]

(e)

(f)

Fig. 10. Similar to Fig. 7a-e and h for [Chl_a] = 0.2 mg/m$^3$ but with additional systematic error of +4% (open squares) and -4% (closed squares) included in the truth-in/truth-out retrieval tests.

[Figure]

[Figure]

(a)

[Figure]

[Figure]

(b)

[Figure]

                                                                                (c)

Fig. 11. (a) Nadir AirMSPI intensity image from spectral combination of 445, 555 and 660 nm bands (left image)

and DoLP image from spectral combination of 470, 660 and 865 nm bands (right image). The bright spot inside the white circle marked on the intensity image (dark spot inside the white circle marked on the DoLP image) is the

AERONET USC_SeaPRISM ocean color station, located on the Eureka oil platform. AirMSPI observations were acquired at 19:44 UTC on 6 February 2013. The yellow frame bounds the area viewed in common from all nine angles (observations over this area are used for retrieval); (b) Maps of retrieved AOD at 555 nm (top left), SSA at

555 nm (top right), nLw at 445 nm (bottom left), and nLw at 555 nm (bottom right); (c) Comparisons of retrieved spectral AOD (top left), SSA (top right), aerosol size distribution (bottom left), and nLw (bottom right) using the

GRASP and JPL algorithms to AERONET reported values.

[Figure]

[Figure]

(a)

[Figure]

[Figure]

[Figure]

(b)

[Figure]

                                                 (c)

Fig. 12. Similar to Fig. 11 but corresponding to the AirMSPI observations near AERONET La Jolla site on 14

January 2013 at 21:09 UTC. The bluish part at the bottom right part of the DoLP image indicates the shallow water area which was not captured by all images and hence excluded in retrieval.